# Long-term dietary nitrate supplementation does not reduce renal cyst growth in experimental autosomal dominant polycystic kidney disease

Jennifer Q. J. Zhang[1,2], Sayanthooran Saravanabavan[1,2], Kai Man Cheng[1,2], Aarya Raghubanshi[1,2], Ashley N. Chandra[1,2], Alexandra Munt[1,2], Benjamin Rayner[3], Yunjia Zhang[3], Katrina Chau[4], Annette T. Y. Wong[1,2], Gopala K. Rangan[1,2]*

1 Centre for Transplant and Renal Research, Westmead Institute for Medical Research, The University of Sydney, Sydney, New South Wales, Australia, 2 Department of Renal Medicine, Westmead Hospital, Western Sydney Local Health District, Sydney, New South Wales, Australia, 3 Heart Research Institute, Sydney Medical School, The University of Sydney, Sydney, New South Wales, Australia, 4 Department of Renal Medicine and School of Medicine, Western Sydney University at Blacktown Hospital, Sydney, New South Wales, Australia

* g.rangan@sydney.edu.au

**Data Availability Statement:** All relevant data are within the paper and its Supporting information files.

## Abstract

Augmentation of endogenous nitric oxide (NO) synthesis, either by the classical L-arginine-NO synthase pathway, or the recently discovered entero-salivary nitrate-nitrite-NO system, may slow the progression of autosomal dominant polycystic kidney disease (ADPKD). To test this hypothesis, the expression of NO in human ADPKD cell lines (WT 9–7, WT 9–12), and the effect of L-arginine on an *in vitro* model of three-dimensional cyst growth using MDCK cells, was examined. In addition, groups of homozygous $Pkd1^{RC/RC}$ mice (a hypomorphic genetic ortholog of ADPKD) received either low, moderate or high dose sodium nitrate (0.1, 1 or 10 mmol/kg/day), or sodium chloride (vehicle; 10 mmol/kg/day), supplemented drinking water from postnatal month 1 to 9 (n = 12 per group). *In vitro*, intracellular NO, as assessed by DAF-2/DA fluorescence, was reduced by >70% in human ADPKD cell lines, and L-arginine and the NO donor, sodium nitroprusside, both attenuated *in vitro* cyst growth by up to 18%. In contrast, in $Pkd1^{RC/RC}$ mice, sodium nitrate supplementation increased serum nitrate/nitrite levels by ~25-fold in the high dose group (P<0.001), but kidney enlargement and percentage cyst area was not altered, regardless of dose. In conclusion, L-arginine has mild direct efficacy on reducing renal cyst growth *in vitro*, whereas long-term sodium nitrate supplementation was ineffective *in vivo*. These data suggest that the bioconversion of dietary nitrate to NO by the entero-salivary pathway may not be sufficient to influence the progression of renal cyst growth in ADPKD.

**Funding:** The research was funded by the National Health and Medical Research Council of Australia (Project Grant Nos. 1164128 and 1138533) and PKD Australia (Recipient G.R.). J.Z. is supported by a Research Training Program Stipend from the University of Sydney. The funders had no role in study design, data collection and analysis, decision to publish, or preparation of the manuscript.

**Competing interests:** The authors have declared that no competing interests exist.

## Introduction

Autosomal dominant polycystic kidney disease (ADPKD) has a population prevalence of 1:1000 and is the most common monogenic cause of kidney failure in adults [1, 2]. It is caused by loss-of-function variants in either *PKD1* or *PKD2*, which encode the trans-membranous, polycystin-1 and 2 proteins, respectively [3]. The formation of expansile cysts which compress surrounding healthy renal parenchyma leads to late-onset kidney failure, and this, together with cardiovascular disease (hypertension, heart and valvular abnormalities and vascular aneurysms), results in premature death in patients with ADPKD [3–5].

Accumulating evidence indicates that the loss of polycystin-1, together with oxidative stress, impairs nitric oxide synthase (NOS) activity and the endogenous synthesis of nitric oxide (NO) in ADPKD [6–10]. In non-genetic ortholog rat models of PKD, the endothelial NOS (eNOS), inducible NOS (iNOS) and neuronal NOS (nNOS) proteins were downregulated in cyst-lining epithelial cells [10, 11]. In the Han:SPRD Cy rat model, L-arginine (a substrate for NOS *via* the well-established L-arginine-NOS pathway [12]) increased NO metabolites, but the therapeutic efficacy on cystic kidney disease was mild [13], possibly due to first-pass metabolism [14]. In contrast, treatment with N(G)-nitro-L-arginine methyl ester (L-NAME, a pan-NOS inhibitor) had more consistent findings and resulted in exacerbation of cystic kidney disease and hypertension [13, 15].

Endogenous NO can also be generated *via* the recently discovered entero-salivary pathway in which the consumption of dietary nitrate is reduced to nitrite by anaerobic bacteria on the dorsal tongue, and then converted to NO [16, 17]. In humans, ~85% of food-derived nitrate is from vegetables (especially red beetroot, spinach, rocket and lettuce, which have the highest content), and the remainder is from variable amounts in drinking water [17–19]. Several studies have investigated the effects of dietary nitrate on augmenting NO [20, 21]. In athero-sclerosis-prone mice, sodium nitrate ($NaNO_3$) in drinking water for 10 weeks restored NO deficiency, and 0.1 mmol/kg/day $NaNO_3$ (equivalent to an additional serving of high nitrate-containing leafy green vegetables in the human diet) increased plasma nitrate by three-fold and improved endothelial function [21]. Furthermore, nitrate-containing beetroot juice increased plasma nitrate and reduced blood pressure (BP) in patients with chronic kidney disease [22].

In this study, we hypothesized that augmentation of NO synthesis, by either the classical L-arginine-NO synthase or the entero-salivary nitrate-nitrite-NO pathway, reduces kidney cyst growth in ADPKD. Previous studies have not directly measured intracellular levels of NO [23], or established the direct effects of L-arginine and NO on cyst growth [13], or assessed the potential long-term adverse effects of augmenting NO, such as increased peroxynitrite formation and/or DNA damage [24]. The aims of this study were to: (i) verify that NO levels in human ADPKD cells are reduced, using a direct method of measurement; (ii) clarify whether L-arginine (NO substrate) and sodium nitroprusside (SNP; NO donor) have direct effects on cyst growth *in vitro* and define their magnitude of therapeutic efficacy; and lastly, (iii) determine the long-term effects of dietary nitrate supplementation on renal cyst growth and markers of DNA damage in $Pkd1^{RC/RC}$ mice (a genetic ortholog of ADPKD).

## Materials and methods

### Cell lines

Four cell lines were obtained from the American Type Culture Collection (ATCC, Manassas VA USA): (i) HK-2; immortalized cells derived from proximal tubules of normal human kidney cortex (CRL-2190, Lot no. 61218770) [25]; (ii) WT 9–7 and; (iii) WT 9–12, both

immortalized cells derived from human ADPKD cysts (CRL-2830, Lot no. 58737172 and CRL-2833, Lot no. 60336584) [26], and; (iv) Madin-Darby canine kidney (MDCK) epithelial cells, a normal kidney epithelial cell line from *Canis familiaris*. HK-2 and MDCK cells were cultured in a 1:1 ratio of DMEM and Ham's F12 with 10% fetal bovine serum (FBS). WT 9–7 and WT 9–12 cells were cultured in DMEM with 10% FBS. All cultures were maintained at 37 ˚C in 5% $CO_2$.

## Assessment of NO levels in human ADPKD cell lines

Diaminofluorescein-2/diacetate (DAF-2/DA), a membrane-permeable, fluorescent NO indicator [27], was used to detect NO levels in human ADPKD cell lines using flow cytometry (fluorescence-activated cell sorter; FACS). DAF-2/DA interacts with the oxidation product of NO, $N_2O_3$, to produce DAF-2T (triazolofluorescein), causing the cells to fluoresce [28]. HK-2, WT 9–7 and WT 9–12 cells (n = 3 per cell type) were seeded in 6-well plates (2 x $10^5$ cells/well) and cultured for two days to confluency. Cells were then washed with PBS and starved overnight in serum-free media. DAF-2/DA (2 μM) was added to the wells and incubated for 30 min at 37 ˚C, followed by fixation in 4% formaldehyde for 15 minutes at room temperature. A FACS analyzer (BD FACSCanto™ II, BD Biosciences, Franklin Lakes, NJ USA) was used to quantify fluorescence (excitation wavelength: 488 nm; and emission wavelength: 530 nm) at the single-cell level, and data were analyzed using FlowJo™ (v10.6.1, BD Biosciences). A total of ~33,000 events/replicate were acquired, and non-cellular particles and debris were excluded by forward and side scatter gating. Final gated cell populations contained ~30,000 cells.

## *In vitro* model of kidney cyst growth

The three-dimensional (3D) model of kidney cyst growth using MDCK cells, whereby cyst growth is induced with forskolin [a cyclic adenosine monophosphate (cAMP) agonist], is a well-established *in vitro* model used to evaluate efficacy of interventions in PKDs [29]. The 3D cyst model was performed as previously described [30]. Briefly, MDCK cells (300 cells/well) were suspended in 0.2 mg/mL ice-cold collagen bovine Type I, supplemented with 10% 10X minimum essential medium, 10 mM HEPES, 27 mM $NaHCO_3$, and 0.2 mM NaOH in 96-well plates. Plates were incubated at 37 ˚C for 90 minutes in a water bath to allow gelation of collagen. Then, culture media containing 10 μM forskolin was added to induce cystogenesis, and cyst formation was observed using a microscope over 8 days.

For the experiments, the culture medium contained either no vehicle (forskolin only), vehicle (0.5% DMSO/2% PBS), L-arginine (0.5, 1 and 2 mM), SNP (5, 10 and 20 μM) or L-NAME (0.25, 0.5 and 1 mM), from day 0. Concentrations were determined based on previous studies using MDCK cells [31], and media was changed every 48 hours. Each 96-well plate had n = 8 replicates for each treatment condition and the experiment was repeated twice. On both day 4 and day 8, at least 10 cyst images were randomly captured in each well, and the cyst diameter was measured by ImageJ (v1.8.0, National Institutes of Health, Bethesda, MD USA) [32]. The diameter of 148–160 cysts were measured for each treatment condition.

## Murine model of ADPKD

*Pkd1*[RC/RC] C57BL/6JAusb mice are an inbred knock-in of a *PKD1* hypomorphic allele (*PKD1 p.R3277C*) [33]. The colony was a kind gift from Dr. Peter Harris (Mayo Clinic, Rochester, MN USA), maintained at Australian BioResources (Moss Vale, NSW Australia), and housed at the Westmead Institute for Medical Research (Westmead, NSW Australia). Mice were allowed food and water *ad libitum* under standard conditions (temperature: 21±2 ˚C; humidity: 55 ±15%; artificial lighting; light: dark cycle 1900–0700). All protocols and procedures were

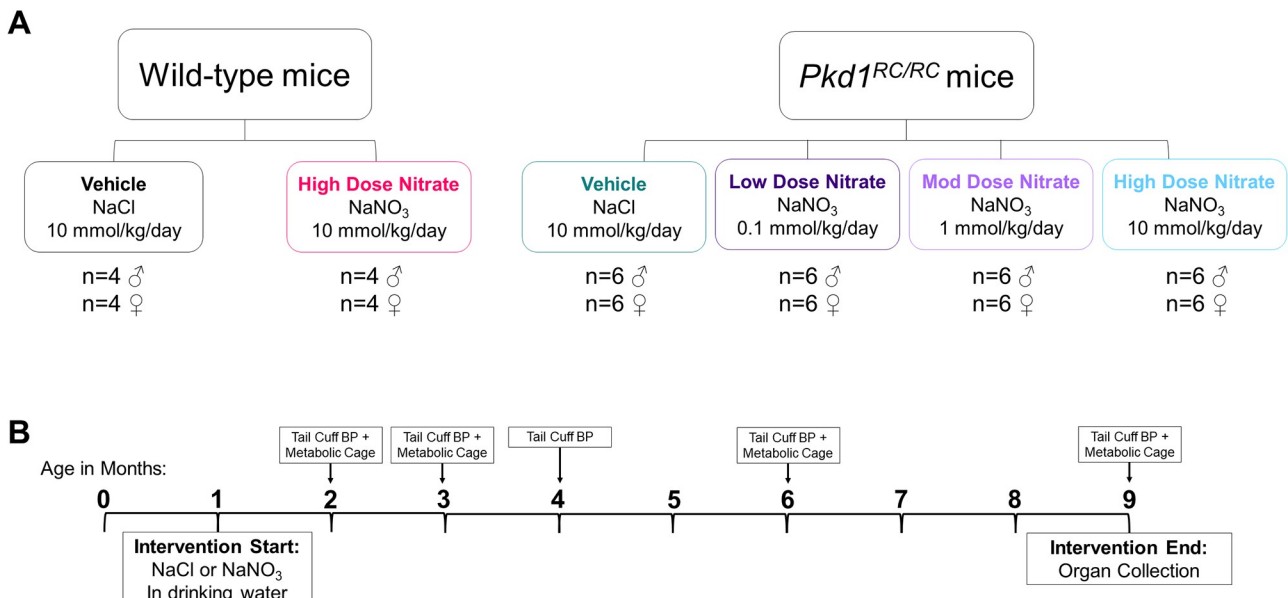

**Fig 1. Experimental design of the animal study.** (A) Experimental groups consisted of wild-type and $Pkd1^{RC/RC}$ mice treated with sodium chloride (NaCl; vehicle) or sodium nitrate (NaNO$_3$; low, 0.1 mmol/kg/day; moderate, 1 mmol/kg/day; high dose, 10 mmol/kg/day); (B) Timeline of the study. Mice were treated with the intervention from 1 to 9 months of age. At 2, 3, 4, 6 and 9 months of age, tail cuff blood pressure (BP) was measured and mice were placed in metabolic cages for urine collection and measurement of fluid intake and urine volume.

approved by the Western Sydney Local Health District Animal Ethics Committee (Protocol numbers 5134 and 5157).

The experimental design for the animal study is shown in Fig 1. Groups of $Pkd1^{RC/RC}$ mice [total n = 12 per group; (6 females and 6 males)] and wild-type mice [total n = 8 per group; (4 females and 4 males)] were randomly allocated to receive either: (i) low dose NaNO$_3$ (0.1 mmol/kg/day), (ii) moderate dose NaNO$_3$ (1 mmol/kg/day), or (iii) high dose NaNO$_3$ (10 mmol/kg/day), supplemented ultrapure (MilliQ®) water from 1 month until 9 months of age [21]. The doses for NaNO$_3$ were based on previous preclinical studies and on the assumption that mice consumed ~6 mL water per day [21, 34]. A range of NaNO$_3$ doses were investigated as high dose NaNO$_3$ was previously found to be ineffective in some disease models due to possible negative cross-talk between endogenous and exogenous NO-generating pathways [21, 35]. Vehicle-treated mice received drinking water supplemented with sodium chloride (NaCl; 10 mmol/kg/day), as described previously [21]. Both NaNO$_3$ and NaCl ($\geq$99.0% purity) were purchased from Sigma Aldrich (St. Louis, MO USA), and were dissolved in ultrapure water to exclude confounding by nitrate/nitrite content present in ordinary tap water [19]. The supplemented drinking water was prepared and changed weekly.

Body weight, fluid intake per cage and food intake per cage were monitored weekly. At 2, 3, 4, 6 and 9 months of age, tail cuff BP was measured and mice were placed in metabolic cages for urine collection and measurement of fluid intake and urine volume. At 9 months of age (8 months of intervention), mice were anaesthetized by intraperitoneal injection of 20% ketamine: 10% xylazine, a mid-line laparotomy was performed, blood was collected *via* cardiac puncture, and organs (kidney, heart and aorta) were removed. The total duration of the intervention was based on the known natural history of renal disease progression of this model [33].

## Analysis of NO metabolites in drinking water and urine/serum

The analysis of nitrate and nitrite content in supplemented drinking water and its stability over seven days was performed by Sydney Analytical Laboratories (Seven Hills, NSW Australia). The analysis of serum and urinary nitrate/nitrite was performed using a commercially available kit (Nitric Oxide Assay Kit, ab65328, Abcam, Cambridge UK), according to manufacturer's instructions.

## Western blot for eNOS and iNOS in mice

Protein lysates were prepared using radioimmunoprecipitation assay (RIPA) buffer with protease (P8340; Sigma-Aldrich) and phosphatase inhibitors (4906845001; Roche, Basel, Switzerland). Sixty-five micrograms was electrophoresed on a 4–15% Mini-Protean® TGX Stain-Free™ precast gel (Bio-Rad Laboratories, Hercules, CA, USA) and transferred to a 0.45 μm PVDF membrane (Bio-Rad Laboratories) by wet transfer. The membrane was then blocked [5% bovine serum albumin (BSA) diluted in Tris-buffered saline with 0.1% Tween (TBST)] for 1 hour, followed by overnight incubation at 4 ˚C with the primary antibody. Primary antibodies used were: (i) anti-iNOS (1:1000, NB300-605; Novus Biologicals, Littleton, CO USA); and (ii) anti-eNOS (1:1000, PA1-037; Invitrogen). The membrane was then incubated with the corresponding secondary antibody (infrared fluorescent-conjugate) for 1 hour at room temperature. Blots were scanned on the Odyssey Imaging System (LI-COR Biosciences, Lincoln, NE USA). Densitometry was quantified using Image Studio Lite (v5.2.5, LI-COR Biosciences) and ImageLab™ Software (v6.0.1, Bio-Rad Laboratories), and normalized against total protein through stain-free imaging technology (Bio-Rad Laboratories) [36, 37].

## Assessment of cystic kidney disease in mice

Coronal slices of the kidney, heart and aorta were immersion-fixed (either in 10% formalin or methyl Carnoy's solution) and paraffin-embedded. Four μm thick tissue sections were cut and stained with Periodic Acid Schiff (PAS) and 0.1% Sirius Red and 0.1% Fast green in Picric acid. For immunohistochemistry, sections were deparaffinized and blocked with 3% hydrogen peroxide. For formalin-fixed slides, antigen retrieval was performed by Decloaking Chamber™ NxGen (95 ˚C for 40 minutes; Biocare Medical, Pacheco, CA USA) in 1X Antigen Decloaker (Biocare Medical). Sections were blocked with Background Sniper (Biocare Medical), and incubated with primary antibodies overnight at 4 ˚C. Primary antibodies used were: (i) anti-iNOS (1:50, NB300-605; Novus Biologicals); (ii) anti-eNOS (1:100, PA1-037; Invitrogen, Carlsbad, CA USA); (iii) anti-mouse F4/80 (1:150, MCA497; Bio-Rad Laboratories); (iv) anti-phospho-histone H2A.X (Ser139) (1:480, 9718; Cell Signaling Technology, Danvers, MA USA); and (v) anti-nitrotyrosine (1:500, A-21285; Invitrogen). Secondary biotinylated or HRP-conjugated antibodies (1:200 dilution) were applied for 30 minutes at room temperature. Vectastain ABC reagent (Vector Laboratories, Burlingame, CA USA) was applied for 20 minutes if needed, followed by diaminobenzidine. Sections were counterstained with methyl green. For quantification, whole-slide digital images were acquired using a slide scanner (NanoZoomer v1, Hamamatsu Photonics, Japan) and whole-slide analysis was performed using the positive pixel algorithm on Aperio ImageScope (v11.2.0.780, Leica Biosystems, Wetzlar, Germany). For percentage positive pixels in the kidney and heart, a total of eight 20X fields of view from the kidney cortex or heart were analyzed and averaged for each mouse.

## Assessment of renal function in mice

To assess the impact of sodium nitrate supplementation on renal function, serum urea was measured from blood samples collected at the time of sacrifice by the Institute of Clinical Pathology and Medical Research (ICPMR), Westmead Hospital (Westmead, NSW Australia).

## Measurement of blood pressure in mice

BP was measured at 2, 3, 4, 6 and 9 months of age using the CODA non-invasive BP system (a tail-cuff method, Kent Scientific Corporation, Torrington, CT USA) [38]. Mice were warmed during BP recordings (heating pad: 33–35 ˚C) and each recording session consisted of 15 inflation/deflation cycles, where the first five cycles were "acclimation" cycles and excluded from analysis [39]. A minimum of five accepted measurements of systolic BP, diastolic BP, mean arterial pressure (MAP) and heart rate (HR) were averaged for each mouse.

## Statistics

One mouse died during BP measurement and was excluded from the analysis, and there was no other mortality during the study. All statistical analyses were performed using JMP® Pro (v14.2.0, SAS Institute, Cary, NC USA). Data were reported as means ± standard deviation (SD). Where appropriate, differences between groups were determined by 2-sample Mann Whitney U test, 2-sample independent t-test or one-way analysis of variance (ANOVA), followed by post-hoc analysis with the Tukey-Kramer HSD test. Significance was defined as $P<0.05$.

## Results

### Intracellular NO levels are reduced in human ADPKD cells

The intracellular level of NO, as determined by DAF-2/DA flow cytometry, was markedly reduced by 75% and 73% in unstimulated WT 9–12 and WT 9–7 cells, respectively, compared to normal kidney tubular cells (HK-2) ($P<0.0001$) (Fig 2). There was no difference in DAF-2/DA fluorescence between WT 9–7 and WT 9–12 cells ($P = 0.609$) (Fig 2). These data suggest that the basal levels of endogenous NO synthesis are impaired in cells mutated with *PKD1*.

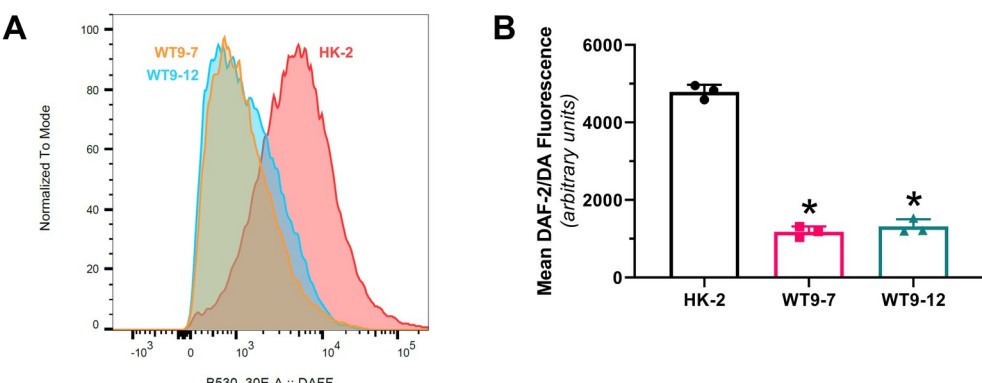

**Fig 2. Fluorescence-activated cell sorting (FACS) analysis for diaminofluorescein-2/diacetate (DAF-2/DA) in unstimulated human ADPKD (WT 9–7 and WT 9–12) and normal kidney (HK-2) cells.** (A) Representative frequency histogram of DAF-2/DA fluorescence intensity; (B) Mean DAF-2/DA fluorescence in WT 9–7 and WT 9–12 cells compared to HK-2. Data presented as means ± SD and represents n = 3 individual experiments. *$P<0.05$ compared to HK-2 control by one-way ANOVA, followed by post-hoc analysis with the Tukey Kramer HSD test.

## NO substrates reduce kidney cyst growth *in vitro*

To evaluate the direct effect of NO modulation, *in vitro* MDCK cysts were treated with either a NOS substrate (L-arginine), a NO donor (SNP) or a NOS inhibitor (L-NAME). Treatment with both L-arginine and SNP attenuated the progression of cyst growth on days 4 and 8 by up to 18%, as shown in Table 1. In contrast, the progression of cyst growth was exacerbated by L-NAME treatment by up to 16% (Table 1). These findings extend previous *in vivo* findings that the modulation of NO alters kidney cyst growth in ADPKD [13].

## Long-term dietary nitrate augments NO metabolites *in vivo*

**Effects of dietary nitrate on general health and levels of NO metabolites.** Treatment with $NaNO_3$ for 8 months had no adverse effects in either wild-type or $Pkd1^{RC/RC}$ mice. The body weight (Fig 3 and S1 Table for body weight sub-analyzed by gender), food intake and fluid intake (measured from cage drinking bottles and metabolic cages) was similar between the groups (S1 Fig). As shown in Table 2, urinary nitrate/nitrite levels at 3 months of age (2 months of intervention) were increased in wild-type and $Pkd1^{RC/RC}$ mice receiving high-dose $NaNO_3$ compared to their corresponding vehicle groups. Furthermore, as expected, urinary nitrate/nitrite was lower in vehicle-treated $Pkd1^{RC/RC}$ mice compared to vehicle-treated wild-type mice (Table 2). Similar to the urine results, serum total nitrate/nitrite levels at 9 months of age (8 months of intervention) were also increased in the high dose $NaNO_3$ groups for wild-type and $Pkd1^{RC/RC}$ mice (97.3 ± 16.3 and 104.6 ± 53.3 nM, respectively) compared to the vehicle groups (0.0 ± 0.6 and 4.0 ± 1.8 nM; P = 0.0002 and P<0.0001, respectively) at the end of the intervention. In contrast, serum nitrate/nitrite in the low and moderate dose $NaNO_3$ groups were not different from the vehicle (4.4 ± 4.5 and 13.9 ± 7.1 nM; P = 1.00 and P = 0.989, respectively).

**Table 1. Effect of L-arginine, SNP and L-NAME on MDCK cyst growth *in vitro*.**

|  | Cyst Diameter *(arbitrary units)* | |
|---|---|---|
|  | **Day 4** | **Day 8** |
| **Forskolin Only** | 1.19 ± 0.22 | 2.53 ± 0.73 |
| **Vehicle (0.5%DMSO+2%PBS)** | 1.13 ± 0.23 | 2.03 ± 0.49 |
| **L-arginine** | | |
| • *0.5 mM* | 1.06 ± 0.22 | 1.82 ± 0.49* |
| • *1 mM* | 1.04 ± 0.29* | 1.68 ± 0.43* |
| • *2 mM* | 1.01 ± 0.20* | 1.69 ± 0.37* |
| **SNP** | | |
| • *5 μM* | 1.07 ± 0.27 | 1.97 ± 0.50 |
| • *10 μM* | 1.05 ± 0.21 | 1.84 ± 0.47* |
| • *20 μM* | 0.94 ± 0.26* | 1.76 ± 0.49* |
| **L-NAME** | | |
| • *0.25 mM* | 1.16 ± 0.24 | 2.15 ± 0.51 |
| • *0.5 mM* | 1.19 ± 0.25 | 2.23 ± 0.59* |
| • *1 mM* | 1.26 ± 0.22* | 2.35 ± 0.60* |

Abbreviations: L-NAME, N(G)-nitro-L-arginine methyl ester; SNP, sodium nitroprusside; DMSO, dimethyl sulfoxide; PBS, phosphate-buffered saline.

Data presented as means ± SD; n = 148–160 cysts were analyzed for each treatment condition.

*P<0.05 compared to same-day Vehicle by one-way ANOVA, followed by post-hoc analysis with the Tukey Kramer HSD test.

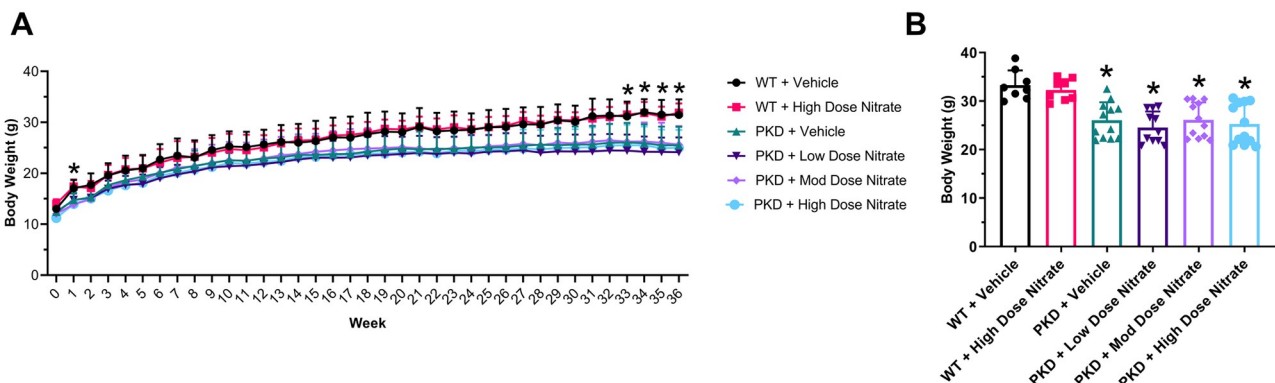

**Fig 3. Body weight in wild-type (WT) and *Pkd1^RC/RC* (PKD) mice treated with sodium chloride (vehicle) or sodium nitrate (low, 0.1 mmol/kg/day; moderate, 1 mmol/kg/day; high dose, 10 mmol/kg/day) for 8 months.** (A) Time-course of body weight. $^*$P<0.05 by one-way ANOVA; (B) Body weight at 9 months of age. $^*$P<0.05 compared to WT + Vehicle by one-way ANOVA, followed by post-hoc analysis with the Tukey Kramer HSD test. Data were combined from male and female mice (n = 4–6 per group per gender) and presented as means ± SD.

**Nitrate and nitrite content in drinking water.**   The nitrate and nitrite content of drinking water provided to experimental mice was analyzed by Sydney Analytical Laboratories (Seven Hills, NSW Australia). As shown in Table 3, ultrapure water and NaCl-supplemented water contained no nitrate or nitrite. Nitrate, but not nitrite, was present in $NaNO_3$-supplemented drinking water, and the nitrate and nitrite content of all water samples remained stable over 7 days (Table 3).

**Expression of iNOS and eNOS in the kidney and aorta.**   Despite the reduction in the levels of NO metabolite in vehicle-treated *Pkd1^RC/RC* mice, and the increase with $NaNO_3$ supplementation, there were no changes in the expression of iNOS and eNOS in the kidney or the aorta between the experimental groups (S2 and S3 Figs). The original uncropped and unadjusted images for the iNOS western blots are shown in S4 Fig.

## Long-term dietary nitrate does not reduce cyst growth *in vivo*

Treatment with $NaNO_3$ did not alter the progression of kidney enlargement or renal function in *Pkd1^RC/RC* mice (Fig 4A and 4B and S2 Table). By histology, *Pkd1^RC/RC* mice developed focal kidney cyst formation, as expected [33], but the percentage kidney cyst area was not altered in

**Table 2. Urinary nitrate/nitrite levels in wild-type (WT) and *Pkd1^RC/RC* (PKD) mice at 3 months of age treated with sodium chloride (vehicle) or sodium nitrate (low, 0.1 mmol/kg/day; moderate, 1 mmol/kg/day; high dose, 10 mmol/kg/day) for 2 months.**

| Group | Urinary Nitrate/Nitrite Level (nM) |
|---|---|
| *WT + Vehicle* | 49.4 ± 19.1 |
| *WT + High Dose Nitrate* | 101.2 ± 28.8$^*$ |
| *PKD + Vehicle* | 16.4 ± 10.8$^*$ |
| *PKD + Low Dose Nitrate* | 33.1 ± 17.9 |
| *PKD + Mod Dose Nitrate* | 48.4 ± 17.9 |
| *PKD + High Dose Nitrate* | 80.6 ± 13.1# |

Data presented as means ± SD (n = 4 per group).

$^*$P<0.05 compared to WT + Vehicle by independent t-test.

#P<0.05 compared to PKD + Vehicle by one-way ANOVA, followed by post-hoc analysis with the Tukey Kramer HSD test.

**Table 3. Analysis of nitrate and nitrite levels in sodium chloride (NaCl) or sodium nitrate (NaNO$_3$) supplemented drinking water fed to wild-type and *Pkd1*$^{RC/RC}$ mice when prepared fresh (day 1) and after 1 week (day 7).**

| Sample | Nitrate (NO$_3^-$) (mg/L) | Nitrite (NO$_2^-$) (mg/L) |
|---|---|---|
| **Day 1** | | |
| *Ultrapure Water Only* | <0.1 | <0.1 |
| *Vehicle* (Ultrapure Water + NaCl) | <0.1 | <0.1 |
| *High Dose Nitrate* (Ultrapure Water + NaNO$_3$) | 2540 ± 17* | <0.1 |
| **Day 7** | | |
| *Ultrapure Water Only* | <0.1 | <0.1 |
| *Vehicle* (Ultrapure Water + NaCl) | <0.1 | <0.1 |
| *High Dose Nitrate* (Ultrapure Water + NaNO$_3$) | 2543 ± 42* | <0.1 |

Data presented as means ± SD (n = 3 per group).

*P<0.05 compared to same-day Ultrapure Water Only and Ultrapure Water + NaCl by one-way ANOVA, followed by post-hoc analysis with the Tukey Kramer HSD test.

*Pkd1*$^{RC/RC}$ receiving NaNO$_3$ supplementation compared to vehicle (Fig 4C and 4D), and this finding was consistent when sub-analyzed by gender (S1 Table). Similarly, interstitial inflammation, measured by F4/80, was increased in *Pkd1*$^{RC/RC}$ mice compared to wild-type but not affected by NaNO$_3$ treatment (Fig 5). Finally, the cardiovascular disease phenotype (tail cuff systolic BP, cardiac enlargement and fibrosis) in vehicle-treated *Pkd1*$^{RC/RC}$ mice was similar to vehicle-treated wild-type mice, preventing the determination of the efficacy of nitrates on improving this disease outcome (S5 and S6 Figs).

## Long-term dietary nitrate does not exacerbate DNA damage accumulation *in vivo*

Because excess NaNO$_3$ may cause undesirable mutagenic effects [40–42], we also evaluated the expression of nitrotyrosine (marker of oxidative stress [43]) and gamma (γ)-H2AX (marker of DNA damage [44]) in the experimental groups. As shown in Fig 6A–6C, there were no differences in the expression of nitrotyrosine in the kidney or aorta of wild-type or *Pkd1*$^{RC/RC}$ mice treated with long-term NaNO$_3$. Consistent with previous findings, γ-H2AX was localized to cyst-lining epithelial cells and peri-cystic areas in *Pkd1*$^{RC/RC}$ mice, but this was also not different between vehicle and NaNO$_3$ groups (Fig 6A & 6D).

## Discussion

Evidence accumulated over the last 20 years has shown that the systemic bioavailability of NO is impaired in ADPKD. Apart from its known effects in mediating endothelial dysfunction, hypertension, and exercise capacity, it has also been shown to mediate kidney cyst growth [13, 23, 45–47]. In this study, we investigated this hypothesis further, and identified three novel findings: (i) *firstly*, intracellular levels of NO, using a direct method of measurement, was markedly reduced in human ADPKD cell lines; (ii) *secondly*, L-arginine and sodium nitroprusside reduced cyst diameter *in vitro*, confirming that modulation of NO has direct effects on cyst growth; (iii) *thirdly*, despite previous evidence supporting the beneficial effects of dietary nitrates in non-renal chronic disease models [20, 21, 48], the continuous administration of sodium nitrate for 8 months, regardless of dosage, did not alter the progression of cyst growth in *Pkd1*$^{RC/RC}$ mice.

Previous studies concluding that the endogenous synthesis of NO is decreased in ADPKD used indirect methods of NO measurement, demonstrating that either endothelium-dependent

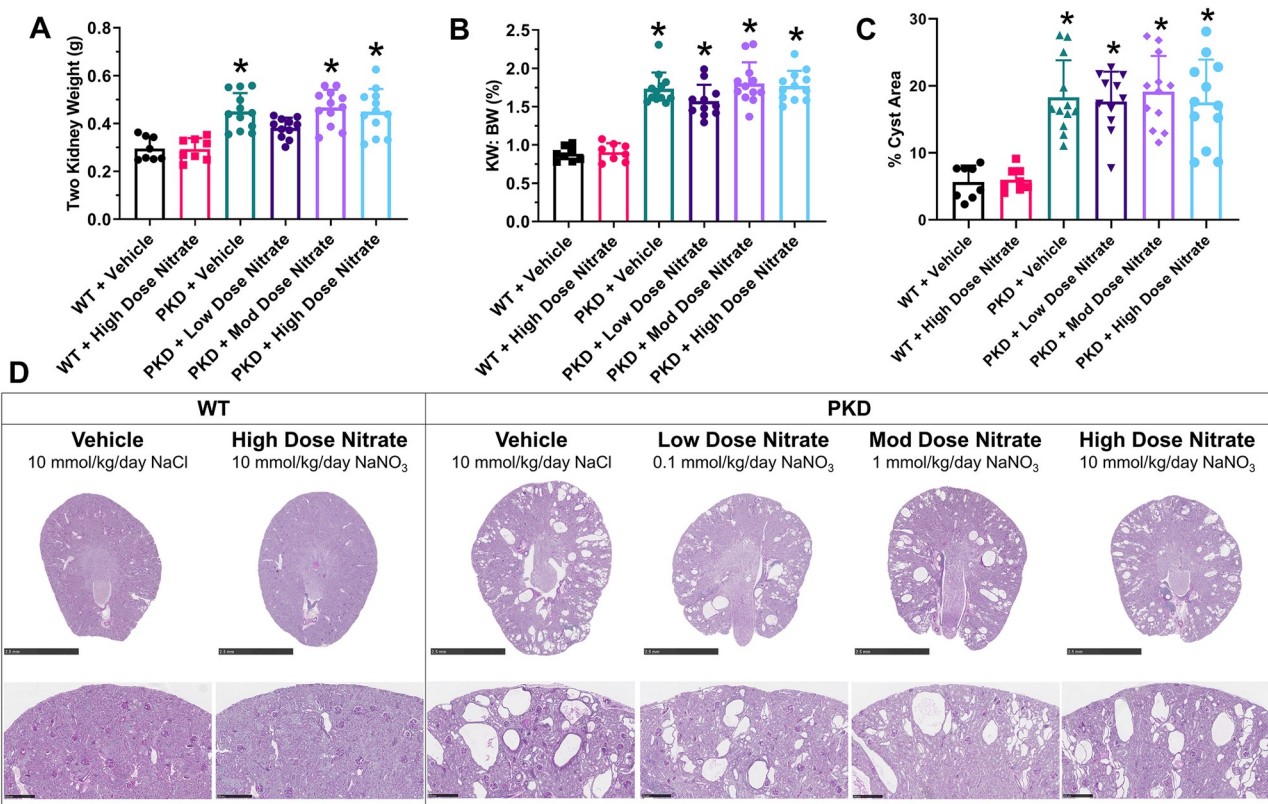

**Fig 4. Kidney enlargement and histological analysis of cystic kidney disease in wild-type (WT) and *Pkd1*[RC/RC] (PKD) mice treated with sodium chloride (vehicle) or sodium nitrate (low, 0.1 mmol/kg/day; moderate, 1 mmol/kg/day; high dose, 10 mmol/kg/day) for 8 months.** (A) Two kidney weight; (B) Kidney weight to body weight ratio (KW: BW); (C) Percentage cyst area; (D) Representative whole-slide digital images of Periodic Acid Schiff (PAS)-stained kidney sections from male WT and PKD mice. *$P < 0.05$ compared to WT + Vehicle by one-way ANOVA, followed by post-hoc analysis with the Tukey Kramer HSD test. Data were combined from male and female mice (n = 4–6 per group per gender) and presented as means ± SD.

relaxation in small resistance vessels or brachial artery flow-mediated dilatation was reduced [23, 49]; and/or that levels of serum/urinary NO metabolite, renal NOS protein/mRNA, and/or NOS activity were reduced [10, 11, 13, 15, 23]. However, baseline levels of NO metabolites and NOS protein have not been consistently reduced in all studies in PKD [50]. These discrepancies highlight the multiple factors that regulate NO *in vivo*, including: the reduction in NO clearance in renal disease [11]; the elevation of naturally occurring NOS inhibitors in ADPKD (such as asymmetric dimethyl arginine) [51–53]; the genetic background of an individual [54]; and/or the need for functional stimulants to uncover the deficit [45]. It is well known that the direct measurement of NO is difficult and complex due to its reactivity and short physical half-life [55, 56]. Therefore, in this study we measured the intracellular levels of bioactive NO in ADPKD using the DAF-2/DA method. DAF-2/DA is a membrane-permeable probe that interacts with $N_2O_3$ (oxidation product of NO) as well as reactive oxygen species, to produce DAF-2T (triazolofluorescein) causing the cells to fluoresce [27, 28, 56]. Our findings showed that DAF-2/DA levels were markedly reduced in human ADPKD cells compared to the control cells, supporting the study hypothesis. While the DAF-2/DA method should be used in combination with other methods [56], its simplicity makes it a useful assay for screening the efficacy of NO modulators in mutated PKD cells *in vitro* and/or cells from patients treated with alternative NO-dependent interventions in future studies [57].

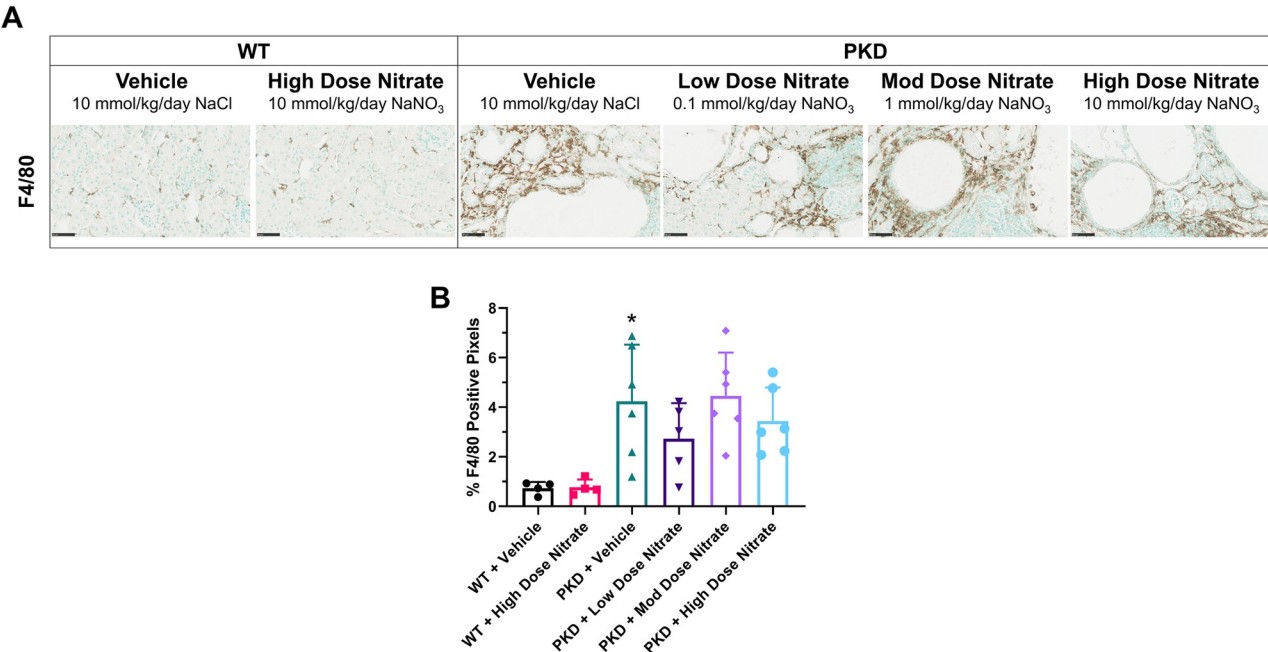

**Fig 5. Interstitial inflammation (F4/80) measured by immunohistochemistry, in wild-type (WT) and *Pkd1*[RC/RC] (PKD) mice treated with sodium chloride (vehicle) or sodium nitrate (low, 0.1 mmol/kg/day; moderate, 1 mmol/kg/day; high dose, 10 mmol/kg/day) for 8 months.** (A) Representative images of positive F4/80 staining; (B) Quantification of F4/80 positive staining. Data presented as means ± SD (n = 4–6 per group). *P<0.05 compared to WT + Vehicle by independent t-test.

The endogenous synthesis of NO is primarily derived from enzymatic, and to lesser extent, non-enzymatic pathways [12, 16]. An increase in the availability of dietary arginine enhances the activity of NOS and promotes NO production [12, 58, 59], and the efficacy of this intervention has been extensively investigated in experimental models of renal disease [60, 61]. However, previous *in vivo* studies in Han:SPRD PKD rats demonstrated that L-arginine only had marginal benefits on reducing kidney weight and cyst volume in males with no effect on BP [13]. This modest effect may be because of the low dose used (0.5 g/L) [13], and/or reduced bioavailability due to first pass metabolism [14]. To further clarify efficacy, in this study, we compared the effects of L-arginine and SNP *in vitro*, and found that they both reduced cyst diameter whereas L-NAME exacerbated it, and interestingly, by similar magnitudes. These data provide evidence that NO has direct cyst-reducing effects. The mechanisms underlying this requires further investigation, but it raises the possibility that NO may have anti-mitogenic properties on cystic epithelial cell proliferation rather than cytotoxicity secondary to free radical formation, as shown previously in mesangial cells [62].

In the present study, we investigated whether the exogenous production of NO from dietary nitrates *via* the recently discovered entero-salivary pathway, will be beneficial in ADPKD [16, 63]. As prelude to the analysis of this study we found that, as expected, urinary levels of NO metabolites were reduced, but the protein expression of iNOS and eNOS in aorta and kidney were not altered in *Pkd1*[RC/RC] mice. The reasons for these disparate findings are not clear, but we did not measure constitutive NOS activity, and this could explain the reduction in NO metabolites despite no significant change in NOS protein levels [12]. Moreover, NOS activity are tissue and isoform-specific, and there may also be alternative mechanisms for the reduction in NO in ADPKD, as described above [11, 45, 51–54].

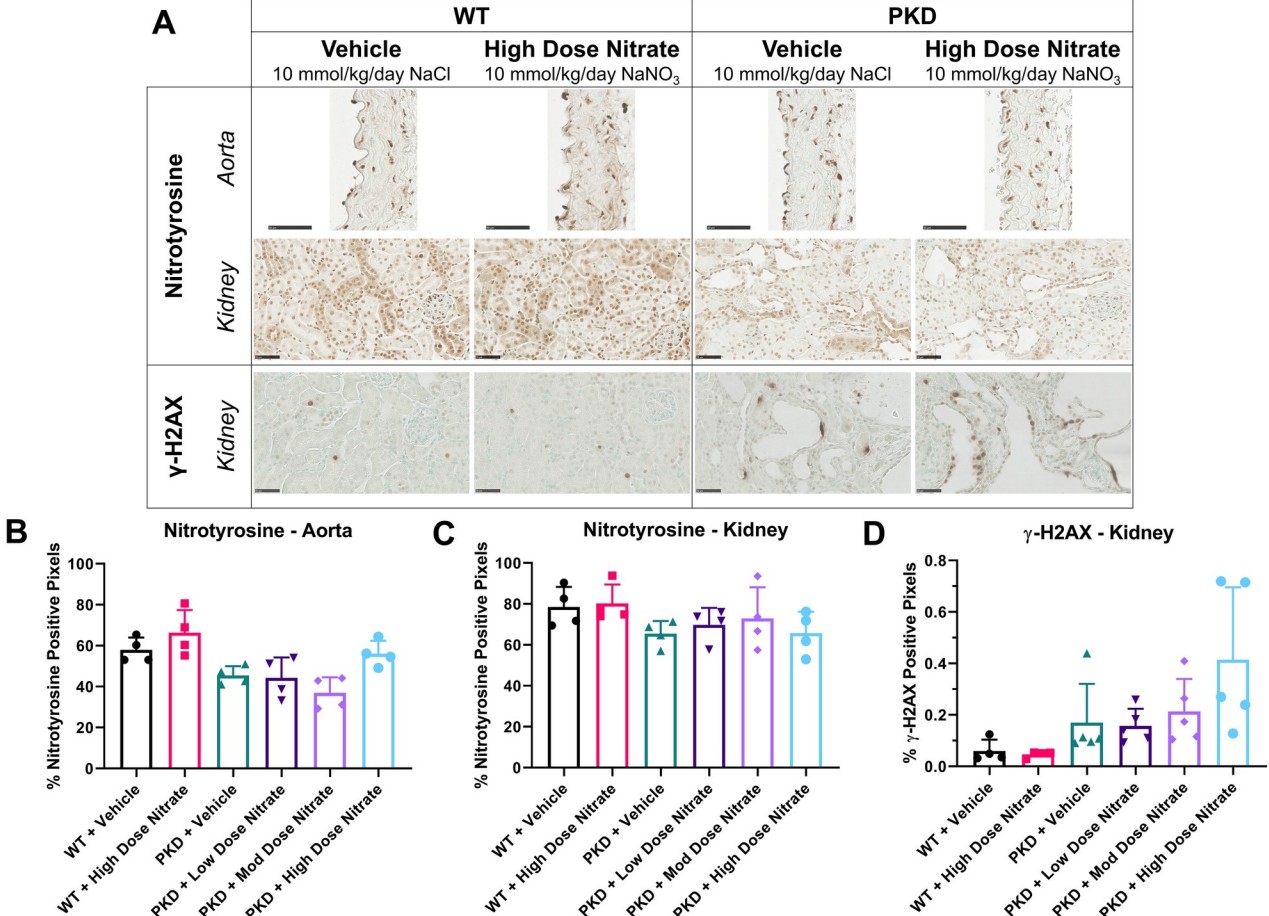

**Fig 6. Immunohistochemistry for nitrotyrosine and gamma (γ)-H2AX in the aorta and kidney of wild-type (WT) and *Pkd1*^RC/RC (PKD) mice treated with sodium chloride (vehicle) or sodium nitrate (low, 0.1 mmol/kg/day; moderate, 1 mmol/kg/day; high dose, 10 mmol/kg/day) for 8 months.** (A) Representative images of nitrotyrosine and γ-H2AX positive staining in the aorta and kidney; (B) Quantification of nitrotyrosine positive staining in the aorta; (C) Quantification of nitrotyrosine positive staining in the kidney; (D) Quantification of γ-H2AX positive staining in the kidney. Data presented as means ± SD (n = 4–6 per group).

Our results showed that despite testing three different doses, and conducting an eight-month follow-up, the progression of kidney cyst growth in *Pkd1*^RC/RC mice was not altered by sodium nitrate supplementation. While there are several possibilities for this lack of efficacy, one simple explanation is that the bioconversion from nitrate to NO is insufficient to alter the renal progression of ADPKD. In support of this possibility, in humans, ~75% of dietary nitrate is excreted by the kidneys, and only ~5–7% is eventually converted to NO by the entero-salivary pathway [16, 64, 65]. Another reason for the lack of efficacy is that negative crosstalk between endogenous and exogenous NO-generating pathways could neutralize the therapeutic effects of sodium nitrate in a dose-dependent manner, and this was the reason we evaluated the effects of three different doses in our study [35]. Negative cross-talk between the endogenous and exogenous NO-generating pathways may exacerbate the impaired net NO bioavailability in ADPKD, and the transient elevation in BP in wild-type mice receiving high dose sodium nitrate at 4–6 months of age suggests that this might be a contributing factor for the lack of efficacy. Lastly, we also investigated whether sodium nitrate supplementation

could result in increased oxidative stress in the kidney which would then mitigate any reno-protective effects [40–42], but found no evidence of increased renal nitrotyrosine formation or DNA damage accumulation with dietary nitrate treatment.

There were several limitations of this study. *First*, although serum nitrate/nitrite levels were increased in the nitrate supplemented mouse groups, bioactive levels of NO were not directly measured [24, 66, 67]. *Second*, the conversion of nitrate to NO by the entero-salivary pathway is influenced by the bacterial composition of both the oral cavity and gut but this was not evaluated in the present study [14, 68–70]. In this regard, it is possible that there are species- and/or strain-specific differences between the microbiome of humans and mice held in an artificial laboratory environment that alter the efficiency of NO generation from dietary nitrate [71, 72]. *Third*, an intermittent dosing regimen for oral nitrate could also have reduced the likelihood of negative crosstalk between the two NO-generating pathways in comparison to continuous administration of nitrate, as was done in the current study. Finally, differences between the design of the *in vitro* and *in vivo* studies (NO substrates vs entero-salivary pathways) does not allow the results to be directly compared with each other.

In conclusion, one of the most important findings of the present study was that long-term supplementation of dietary nitrate intake did not alter the progression of cystic kidney disease of a murine genetic ortholog of ADPKD, contrary to our initial hypothesis. The reasons for the lack of efficacy are not fully clear, but suggest that this pathway may be inefficient in generating endogenous NO for disease modification in ADPKD, and/or that negative crosstalk with exogenous NO systems maintains impaired net NO bioavailability. Because the reduced bioavailability of NO is a consistent feature of the ADPKD phenotype and may mediate both renal and cardiovascular disease, further studies are needed to investigate this hypothesis. Perhaps, future approaches could include natural sources of nitrate (such as beetroot juice containing betaine, which has demonstrated disease-modifying effects in preliminary studies using $Pkd1^{RC/RC}$ mice [73, 74]), and/or pharmacological modulators of NO, in particular soluble guanylyl cyclase stimulators (such as taxol, nebivolol and parcigalcut [57, 75, 76]). It is important to gain a better understanding of the role of the NO pathway in ADPKD as it will provide opportunities to develop therapies that could not only attenuate kidney cyst growth, but also reduce complications due to hypertension.

## Supporting information

**S1 Fig. Fluid intake, food intake and urine volume in wild-type (WT) and $Pkd1^{RC/RC}$ (PKD) mice treated with sodium chloride (vehicle) or sodium nitrate (low, 0.1 mmol/kg/ day; moderate, 1 mmol/kg/day; high dose, 10 mmol/kg/day) for 8 months.** (A) Fluid intake by cage bottles measured weekly for 8 months; (B) Fluid intake by metabolic cage measured at 2, 3, 6 and 9 months of age; (C) Urine volume measured at 6 and 9 months of age; (D) Food intake measured weekly for 8 months. Data were combined from male and female mice (n = 4–6 per group per gender) and presented as means ± SD. *P<0.05 by one-way ANOVA.
(TIF)

**S2 Fig. The expression of inducible and endothelial nitric oxide synthase (iNOS and eNOS) in wild-type (WT) and $Pkd1^{RC/RC}$ (PKD) mouse kidneys at 1 and 9 months of age, measured by immunohistochemistry and western blotting.** (A) Representative images of iNOS positive staining in mouse kidneys; (B) Quantification of iNOS positive staining in mouse kidneys; (C) Representative western blots of iNOS in mouse kidneys; (D) Quantification of renal iNOS in mouse kidneys by western blot; (E) Representative images of eNOS

positive staining in mouse kidneys; (F) Quantification of eNOS positive staining in mouse kidneys. Data presented as means ± SD (n = 4–6 per group).
(TIF)

**S3 Fig. The expression of endothelial and inducible nitric oxide synthase (eNOS and iNOS) in the aorta of wild-type (WT) and *Pkd1*^RC/RC (PKD) mice treated with sodium chloride (vehicle) or sodium nitrate (low, 0.1 mmol/kg/day; moderate, 1 mmol/kg/day; high dose, 10 mmol/kg/day) for 8 months, measured by immunohistochemistry.** (A) Representative images of eNOS positive staining in mouse aorta; (B) Quantification of eNOS positive staining in mouse aorta; (C) Representative images of iNOS positive staining in mouse aorta; (D) Quantification of iNOS positive staining in mouse aorta. Data presented as means ± SD (n = 4–6 per group).
(TIF)

**S4 Fig. Original uncropped and unadjusted western blot images of S2C Fig showing the expression of inducible nitric oxide synthase (iNOS) in wild-type (WT) and *Pkd1*^RC/RC (PKD) mouse kidneys at 1 and 9 months of age.** Top panels show immunological detection of iNOS and bottom panels show total protein as obtained by Stain-Free imaging technology (Bio-Rad Laboratories).
(TIF)

**S5 Fig. Blood pressure and heart rate in wild-type (WT) and *Pkd1*^RC/RC (PKD) mice treated with sodium chloride (vehicle) or sodium nitrate (low, 0.1 mmol/kg/day; moderate, 1 mmol/kg/day; high dose, 10 mmol/kg/day) for 8 months.** Panels (A), (B), (C) and (D) show fold-change in systolic (D & H), diastolic (E & I), mean arterial blood pressure (F & J) and heart rate (G & K) at 9 months of age compared to 2 months. *P<0.05 compared to WT + Vehicle by one-way ANOVA, followed by post-hoc analysis with the Tukey Kramer HSD test. (D & H) Time-course of systolic blood pressure; (E & I) Time-course of diastolic blood pressure; (F & J) Time-course of mean arterial pressure; and (G & K) Time-course of heart rate. *P<0.05 by one-way ANOVA. Data were combined from male and female mice (n = 4–6 per group per gender) and presented as means ± SD.
(TIF)

**S6 Fig. Cardiac enlargement and fibrosis (Sirius Red deposition) in the heart and aorta of wild-type (WT) and *Pkd1*^RC/RC (PKD) mice treated with sodium chloride (vehicle) or sodium nitrate (low, 0.1 mmol/kg/day; moderate, 1 mmol/kg/day; high dose, 10 mmol/kg/day) for 8 months.** (A) Heart weight; (B) Heart weight to body weight ratio (HW: BW). Data were combined from male and female mice (n = 4–6 per group per gender) and presented as means ± SD. (C) Quantification of Sirius Red positive staining in mouse heart. (D) Quantification of Sirius Red positive staining in mouse aorta. Data presented as means ± SD (n = 4–6 per group). (E) Representative images of Sirius Red deposition in heart and aorta.
(TIF)

**S1 Table. Body weight, kidney enlargement and percentage cyst area in wild-type (WT) and *Pkd1*^RC/RC (PKD) mice treated with sodium chloride (vehicle) or sodium nitrate (low, 0.1 mmol/kg/day; moderate, 1 mmol/kg/day; high dose, 10 mmol/kg/day) for 8 months, sub-analyzed by gender.**
(DOCX)

**S2 Table. Serum urea in wild-type (WT) and *Pkd1*^RC/RC (PKD) mice treated with sodium chloride (vehicle) or sodium nitrate (low, 0.1 mmol/kg/day; moderate, 1 mmol/kg/day;**

**high dose, 10 mmol/kg/day) for 8 months.**
(DOCX)

## Acknowledgments

Histology, cell imaging, and flow cytometry were performed at the Westmead Scientific Platforms, which are supported by the Westmead Research Hub, the Westmead Institute for Medical Research of Australia, the Cancer Institute New South Wales, the National Health and Medical Research Council and the Ian Potter Foundation.

## Author Contributions

**Conceptualization:** Jennifer Q. J. Zhang, Sayanthooran Saravanabavan, Annette T. Y. Wong, Gopala K. Rangan.

**Data curation:** Jennifer Q. J. Zhang, Sayanthooran Saravanabavan, Gopala K. Rangan.

**Formal analysis:** Jennifer Q. J. Zhang, Sayanthooran Saravanabavan, Kai Man Cheng, Aarya Raghubanshi, Gopala K. Rangan.

**Funding acquisition:** Gopala K. Rangan.

**Investigation:** Jennifer Q. J. Zhang, Sayanthooran Saravanabavan, Kai Man Cheng, Aarya Raghubanshi, Gopala K. Rangan.

**Methodology:** Jennifer Q. J. Zhang, Sayanthooran Saravanabavan, Kai Man Cheng, Aarya Raghubanshi, Benjamin Rayner, Yunjia Zhang, Katrina Chau, Annette T. Y. Wong, Gopala K. Rangan.

**Project administration:** Jennifer Q. J. Zhang, Gopala K. Rangan.

**Resources:** Jennifer Q. J. Zhang, Sayanthooran Saravanabavan, Gopala K. Rangan.

**Supervision:** Sayanthooran Saravanabavan, Annette T. Y. Wong, Gopala K. Rangan.

**Validation:** Jennifer Q. J. Zhang, Sayanthooran Saravanabavan, Gopala K. Rangan.

**Visualization:** Jennifer Q. J. Zhang, Sayanthooran Saravanabavan, Gopala K. Rangan.

**Writing – original draft:** Jennifer Q. J. Zhang, Gopala K. Rangan.

**Writing – review & editing:** Jennifer Q. J. Zhang, Sayanthooran Saravanabavan, Kai Man Cheng, Aarya Raghubanshi, Ashley N. Chandra, Alexandra Munt, Benjamin Rayner, Yunjia Zhang, Katrina Chau, Annette T. Y. Wong, Gopala K. Rangan.

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
