## [Decision Letter · Decision Letter 0]

9 Oct 2020

PONE-D-20-29428

Oral nitrate supplementation does not reduce cardiovascular or cystic kidney disease in murine autosomal dominant polycystic kidney disease

PLOS ONE

Dear Dr. Zhang,

Thank you for submitting your manuscript to PLOS ONE. After careful consideration, we feel that it has merit but does not fully meet PLOS ONE’s publication criteria as it currently stands. Therefore, we invite you to submit a revised version of the manuscript that addresses the points raised during the review process. Three reviewers have read your manuscript. All 3 find it of some interest. however, a number of important issues are raised that ALL  must be addressed in your rebuttal and in the revised manuscript. Please include the following items when submitting your revised manuscript:

We look forward to receiving your revised manuscript.

Kind regards,

Jaap A. Joles, DVM, PhD

Academic Editor

PLOS ONE

Journal Requirements:

2.PLOS ONE now requires that authors provide the original uncropped and unadjusted images underlying all blot or gel results reported in a submission’s figures or Supporting Information files. This policy and the journal’s other requirements for blot/gel reporting and figure preparation are described in detail at https://journals.plos.org/plosone/s/figures#loc-blot-and-gel-reporting-requirements and https://journals.plos.org/plosone/s/figures#loc-preparing-figures-from-image-files. When you submit your revised manuscript, please ensure that your figures adhere fully to these guidelines and provide the original underlying images for all blot or gel data reported in your submission. See the following link for instructions on providing the original image data: https://journals.plos.org/plosone/s/figures#loc-original-images-for-blots-and-gels.

Reviewers' comments:

Reviewer's Responses to Questions

**Comments to the Author**

1. Is the manuscript technically sound, and do the data support the conclusions?

Reviewer #1: No

Reviewer #2: Partly

Reviewer #3: Yes

2. Has the statistical analysis been performed appropriately and rigorously? 

Reviewer #1: Yes

Reviewer #2: Yes

Reviewer #3: Yes

3. Have the authors made all data underlying the findings in their manuscript fully available?

Reviewer #1: Yes

Reviewer #2: Yes

Reviewer #3: Yes

4. Is the manuscript presented in an intelligible fashion and written in standard English?

Reviewer #1: Yes

Reviewer #2: Yes

Reviewer #3: Yes

5. Review Comments to the Author

Reviewer #1: The vascular abnormalities in polycystic kidney disease have attracted significant attention lately. Understanding the mechanisms behind it and the therapeutics to lower their impact will be of significant importance. Prior studies have demonstrated that there is a decrease in the availability of nitric oxide and nitric oxide is seen days. In this study, Zhang et al study the effect of oral nitrates (as a way of nitric oxide donation) on the cardiovascular and cystic kidney disease progression in aim urine model of ADPKD.

From the perspective of the significance, the timing of the study is very appropriate. It is important that they use an intervention that is fairly inexpensive with potential widespread application.

The study uses a well-accepted mouse model of PKD (RCRC). This reviewer agrees with some of the conclusions of the failure of oral nitrate supplementation to “normalize” some of the renal variables.

This reviewer has the following concerns:

1. A number of the vascular variables analyzed do not show any difference between PKD and wild-type. That lack of difference between PKD and WT do not allow making conclusions of nitrate therapy.

2. Interestingly, but not surprising, high-dose nitrates had a “worse outcomes” in some of the variables analyzed like endothelial function. This is likely to the fact that nitric oxide, as mentioned by the others, could have of vasoconstrictor effect in conditions of normal nitric oxide by availability.

3. The authors provide evidence of the decrease in aortic eNOS in PKD compare to wild-type. He will be important to know what happens after oral nitrate supplementation. Authors bring up that point but do not provide explanation why that data is not available.

4. I think that the overall conclusion of the study is over achieving, including the title. I do not think that the data supports that conclusion. It may support some of the findings but not all of them.

5. Minor comment: Please provide significance is on figure legends and in graphics.

Reviewer #2: In this well written manuscript, the authors provide data that chronic sodium nitrate (8 months) in drinking water fails to protect against the progression of polycystic kidneys in a mouse model, Pkd1 RC/RC, despite the fact that ADPKD cell cultures show that NO levels are reduced, and supplementation of MDCK cells with NOS substrate, L-Arg, and SNP, NO donor, modestly reduce cyst growth and L-NAME exacerbates cyst growth.

Suggestions for revision:

1. Introduction, line 79, “serve” should be “serving”.

2. Intro, line 86: “nitrate” should be “nitrates”. The authors make the point in this sentence that there are high levels of nitrates in the typical diet. If this is the case and NO is thought to be protective against PKD, then why does PKD occur? Perhaps the authors should be more circumspect about this.

3. The figures are all “fuzzy” and are difficult to read. Figure 4C—the MWs of the bands cannot be read. Also, was there a loading control used for the western blots? Also, there is no representative eNOS western. Also, the notations for statistical significances in the figures are difficult to see.

4. Was serum/urine creatinine measured by LC/MS which is required for mouse measurements. If not, this is the reason why the data were zero. The authors should consider performing the creatinines by LC/MS or remove them from the paper.

5. Are the authors convinced that the NOx kit measures only NO and not peroxynitrite?

6. Figure 6, why do BPs (Syst, Dias, Mean) increase on high dose nitrate at 4 and 6 months of age, but then decrease at 9 months?

7. The data are not completely unexpected. Other investigators have shown the chronic oral nitrate/nitrite do not reduce blood pressure or protect against renal function. Did the authors measure any index of oxidative stress in the mice—F2-isoprostanes (urine), total antioxidant capacity (serum). If the chronic nitrate did not reduce oxidative stress, then it is not surprising that there was no protection against renal injury or BP. Also, since NO binds with greater affinity to superoxide, then perhaps the authors were not developing a higher concentration of NO compared to the superoxide, to overcome the oxidative stress. In addition, if the nitrate did provide NO for binding to superoxide, then peroxynitrite levels would also increase and studies have shown that chronic peroxyntrite causes vasodilation, but this quickly becomes tachyphylactic and loses potency or even increases BP and renal injury. The authors may want to address this possibility in their Discussion.

Reviewer #3: Overall summary:

This manuscript tested the hypothesis that NO deficiency is feature of ADPKD, and that long-term oral nitrate supplementation reduces cardiovascular and cystic kidney disease progression in a genetic ortholog of ADPKD (Pkd1 RC/RC mice). In their analyses, a human ADPKD tubular cell line, NO was reduced by 75% to 73%, compared to controls. In Pkd1 RC/RC mice, the expression of aortic endothelial NO synthase (NOS) was decreased by 18%. Moreover, in the three-dimensional MDCK in vitro cyst model, treatment with NO substrates (L-arginine and sodium nitroprusside) reduced cyst growth by up to 18%. In Pkd1 RC/RC mice, sodium nitrate supplementation increased serum nitrate/nitrite levels by ~25-fold compared to control, with no adverse effects. However, sodium nitrate supplementation for 8 months, regardless of dosage, did not alter the progression of cardiovascular (assessed by tail cuff systolic blood pressure, ex vivo aortic wire myography and cardiac fibrosis) or cystic kidney disease in Pkd1 RC/RC mice. In conclusion, their data suggested that NO synthesis is reduced in ADPKD, but the long-term intake of oral nitrate in drinking water is not sufficient to correct this deficiency and attenuate cardiovascular and kidney disease progression. Overall, while this article is interesting and could potentially be of significance in ADPKD field, there are significant points that the authors could clarify better. At this stage, this article would benefit from a fair amount of editing before publishing.

Major comment:

1. To test their hypothesis of oral nitrate, to reduce cardiovascular and kidney diseases progression, authors didn’t establish reliable control before executing their animal model experiments. Despite the fact that oral nitrate has been shown to reduce blood pressure (J Appl Physiol (1985). 2019 Oct 1;127(4):1085-1094.), Wild-type (control) treated with high dose of nitrate had an increase blood pressure (Figure 6 D-F). This raises the question of the purity and/or the nitrate supplement that was used in this study. Or, perhaps the (PKD1 RC/RC mouse model) have an impaired microbiota in bioactivation of dietary nitrate (Free Radic Biol Med. 2019 Dec;145:342-348.).

Minor comments:

1. Page 14: Figure 1 need to be mentioned in the result section before start interpreting Figure 2.

2. Page 15, Line 282: Please delete the duplication: “P<0.05 compared to compared to…”

3. Figure 3 and Figure 4: kidney tissue Images are very poor quality that is not easy to asses.

4. Figure 4 E-F: It is strange that the authors have examined the expression of iNOS (4 c-d) and kind of ignored all together the expression of eNOS to validate their immunohistochemistry results.

5. Flow of the figures and results is hard to follow, and need be to more organize. For example, Supplemental figures are not in order throughout the manuscript (S1 vs S2 and so on).

6. Table 1: There is no rational, or nothing was given by authors, is to why they selected MCDK cell line over WT9-7 and/or WT9-12, nor any justification of selected concentrations.

7. Page 17, Line 329: Again please delete the duplication “P<0.05 compared to compared to…”

8. Figure 5: Again, inconsistency of representing the results. Kidney weight and kidney to body weight ratio are given, but not the same with heart. Also, authors in (Fig 5C-E) are over interpreting these data, Page 17, Line 336, “Continuous treatment with NaNO3 for 8 months…did not alter the progression of kidney or heart enlargement in Pkd1RC/RC mice.” For example, in figure 5C low dose of Nitrate seems to have directly reduced the kidney enlargement, despite the KW: BW. In the contrary, moderate dose of Nitrate had at least increased heart to body weight ratio. This further might suggest that Nitrate effect might be dose dependent. Have the authors at least sub-analyzed these data by gender?

9. Page 17, Line 363: confusing statement and perhaps incorrect. “Neither tail-cuff blood pressure nor heart rate were altered by long-term NaNO3 supplementation in wild-type or Pkd1RC/RC mice (Fig 6A-I & S3A-C Fig).” First, Blood pressure of Wild-type had indeed been negatively altered by long-term NaNO3 (Figure 6 D-F), particularly in the 6th month. Second, again authors need to explain that how high dose of nitrate is increasing BP rather than decreasing it in their control. This raises the questions of the purity and/or the supplement that was administered to the mice. Furth, Please move more relevant heart rate supplemental result to the main figure.

6. PLOS authors have the option to publish the peer review history of their article (what does this mean?). If published, this will include your full peer review and any attached files.

Reviewer #1: No

Reviewer #2: No

Reviewer #3: No

---

## [Author Response · Author response to Decision Letter 0]

22 Dec 2020

Response to Reviewers 

Reviewer #1:

Reviewer #1 Comment 1: A number of the vascular variables analyzed do not show any difference between PKD and wild-type. That lack of difference between PKD and WT do not allow making conclusions of nitrate therapy.

Author Response: We agree with Reviewer #1 that the vascular phenotype of the Pkd1RC/RC mouse model is mild. Perhaps, additional interventions, such as a high-salt diet or uninephrectomy, are required to accelerate the vascular phenotype of this model. It is important to note that there are no genetically orthologous models of ADPKD that exhibit measurable long-term cardiovascular disease progression, as vascular defects due to PKD1 and/or PKD2 mutations often lead to embryonic lethality [1-4]. Other experimental models of PKD that exhibit worsening cardiovascular disease over a measurable experimental time period are not genetically orthologous [5, 6], and thus translatability of this intervention is uncertain even if oral nitrate supplementation is shown to be beneficial in these hypertensive PKD models. 

To address this comment, we have made the following changes in the manuscript:

• The Discussion has been revised to include this as a limitation (p27-28, Lines 559-67).

• The Title of the manuscript has been revised (p1, Lines 1-3). 

Reviewer #1 Comment 2: high-dose nitrates had a “worse outcomes” in some of the variables analyzed like endothelial function. This is likely to the fact that nitric oxide, as mentioned by the others, could have of vasoconstrictor effect in conditions of normal nitric oxide by availability.

Author Response: We thank Reviewer #1 for highlighting this point and to address this we have made the following changes to the manuscript: 

• Results (p19, Lines 364-68): Added a comment that high dose NaNO3 transiently increased systolic blood pressure at 4 and 6 months, and also increased diastolic and mean arterial blood pressure at 6 months, in wild-type mice.

• Discussion (p25, Lines 500-09): As mentioned by Reviewer #1, the mechanisms may be due to the negative cross-talk between the endogenous and exogenous NO-generating pathways, leading to downregulation of eNOS activity and an overall reduction in net NO bioavailability, as shown in previous studies [7]. However, in our study we found that total eNOS did not change with nitrate supplementation in wild-type or Pkd1RC/RC mice (see our Response to Comment 3). Therefore, we performed additional experiments to examine whether this elevation in blood pressure may have been related to oxidative stress as peroxynitrite has been associated with vascular dysfunction [8, 9], however, nitrotyrosine expression was not altered with oral nitrate supplementation in wild-type or Pkd1RC/RC mice (see Response to Reviewer #2 Comment 8). Therefore, we agree that the data suggests that high dose nitrate may be associated with worser disease outcomes, but the reasons behind the transiently higher blood pressure in wild-type mice on high dose NaNO3 remain unclear. 

Reviewer #1 Comment 3: The authors provide evidence of the decrease in aortic eNOS in PKD compare to wild-type. It will be important to know what happens after oral nitrate supplementation. 

Author Response: We performed additional experiments to address this question. The results demonstrated that total eNOS expression did not change with oral nitrate supplementation in neither wild-type nor Pkd1RC/RC mice (see Results: p20-21, Lines 400-414 and Fig 7). These data are consistent with previous studies which have also demonstrated no change in total eNOS in response to oral nitrate supplementation [7, 10] (see Discussion: p25, Lines 500-09). 

Reviewer #1 Comment 4: the overall conclusion of the study is overachieving, including the title. I do not think that the data supports that conclusion. It may support some of the findings but not all of them.

Author Response: The following changes have been made to the revised manuscript. 

• The title has been modified to: “Effects of long-term oral nitrate supplementation on the progression of cardiovascular and renal outcomes in murine autosomal dominant polycystic kidney disease” (p1, Lines 1-3). 

• The concluding line of the Abstract has been changed to: “In conclusion, these data confirm that NO synthesis is reduced in ADPKD, but the long-term intake of oral nitrate in drinking water does not alter cardiovascular and cystic kidney disease progression in murine ADPKD.” (p3, Lines 48-50)

• The line, “Thus, these data do not provide supportive evidence to evaluate oral nitrate as a monotherapy in human ADPKD.”, has been removed from the Discussion. 

• We also acknowledged that the lack of efficacy of oral nitrate therapy may be specific to this mouse model and that further studies in other models and using other interventions would be useful, as discussed earlier (see Discussion: p25, Lines 513-19 and p27-28, Lines 559-67). The concluding lines of the manuscript now reinforce that these alternative oral nitrate-related interventions should be examined in preclinical studies to inform future clinical trials in ADPKD (see Discussion: p28, Lines 576-81). 

Reviewer #1 Comment 5: Please provide significance on figure legends and in graphics.

Author Response: Statistical significance has been denoted by * or # on all figures and figure legends depending on the comparison performed. Where relevant, the P-value has also been shown. 

Reviewer #2: 

Reviewer #2 Comment 1: Introduction, line 79, “serve” should be “serving”.

Author Response: This has been corrected (see Introduction: p6, Line 78). 

Reviewer #2 Comment 2: Intro, line 86: “nitrate” should be “nitrates”. The authors make the point in this sentence that there are high levels of nitrates in the typical diet. If this is the case and NO is thought to be protective against PKD, then why does PKD occur? 

Author Response: Thank you for your comment. The majority (85%) of dietary nitrates are derived from vegetables [11, 12]. The consumption of vegetables is low in the Western diet and therefore, the usual intake of dietary nitrates is probably insufficient [13]. Our intention with this sentence was to highlight the potential translatability of the findings should the hypothesis be proven correct. Therefore, to improve the clarity, we have changed line 86 to: “Given the availability of nitrates in natural foods [11], the restoration of NO deficiency by dietary modification via the entero-salivary pathway could be an easily translatable treatment.” (see Introduction: p6, Lines 85-87 in the revised manuscript).

Reviewer #2 Comment 3: The figures are all “fuzzy” and are difficult to read. Figure 4C—the MWs of the bands cannot be read. Also, was there a loading control used for the western blots? Also, there is no representative eNOS western. Also, the notations for statistical significances in the figures are difficult to see.

Author Response: 

• We checked the image quality and noted the PDF proof is compressed and therefore, images can appear “fuzzy”. The images uploaded to the PLOS One server are of high quality. Otherwise, we have increased the font size of labels and notations in all figures as recommended. 

• The western blot data (now Fig 2C-D in the revised manuscript) was normalized to total protein through stain-free imaging technology [14, 15], and thus, a loading control was not necessary. This is described in the Materials and methods (p13, Lines 243-46). 

• Western blot for eNOS was attempted in mouse kidney tissue, but unfortunately did not produce specific bands. As shown in the immunohistochemistry data (Fig 2E in the revised manuscript), eNOS in the mouse kidney was most strongly expressed in endothelial cells which are in low abundance in whole kidney tissue. The eNOS western blot data in kidney tissue has been added as a Supplemental Figure (S2 Fig) and a comment has been added to Results (p15, Lines 291-93). 

Reviewer #2 Comment 4: Was serum/urine creatinine measured by LC/MS which is required for mouse measurements. If not, this is the reason why the data were zero. The authors should consider performing the creatinines by LC/MS or remove them from the paper.

Author Response: Serum creatinine was not measured by LC/MS and thus, this data has been removed from the manuscript. 

Reviewer #2 Comment 5: Are the authors convinced that the NOx kit measures only NO and not peroxynitrite?

Author Response: 

• The method used in the NOx kit is based on the Griess reaction [16]. The Griess reaction, also known as the diazotization assay, converts nitrite (NO2-) to a purple-coloured azo-dye that can be measured by spectrometry [16]. Nitrite reacts with sulfanilamide to form N2O3 which reacts with N-(1-napthyl)ethylenediamine to form the coloured product [17]. To our knowledge, peroxynitrite does not react with sulfanilamide or N-(1-napthyl)ethylenediamine directly, but we acknowledge that Griess reaction is an indirect measure of NO [16], and this has been added to the Discussion as a limitation (p27, Lines 546-49). 

• We performed additional experiments to address the question of peroxynitrite. We measured nitrotyrosine as a surrogate marker for peroxynitrite formation to further explore the adverse effects of oral nitrate supplementation. Please see our response to Comment 8. 

Reviewer #2 Comment 6: Figure 6, why do BPs (Syst, Dias, Mean) increase on high dose nitrate at 4 and 6 months of age, but then decrease at 9 months?

Author Response: We thank Reviewer #2 for highlighting this. Please see our response to Reviewer #1 Comment 2 for an explanation. 

Reviewer #2 Comment 7: The data are not completely unexpected. Other investigators have shown the chronic oral nitrate/nitrite do not reduce blood pressure or protect against renal function. 

Author Response: We agree with Reviewer #2 that the data on the beneficial role of oral nitrate supplementation on blood pressure and chronic kidney disease are uncertain. To our knowledge, this is the first long-term study performed in a murine model of ADPKD, and we hope that the data will be of interest to the PKD community and inform further studies in therapies related to augmenting NO signalling.

Reviewer #2 Comment 8: Did the authors measure any index of oxidative stress in the mice—F2-isoprostanes (urine), total antioxidant capacity (serum). If the chronic nitrate did not reduce oxidative stress, then it is not surprising that there was no protection against renal injury or BP. Also, since NO binds with greater affinity to superoxide, then perhaps the authors were not developing a higher concentration of NO compared to the superoxide, to overcome the oxidative stress. In addition, if the nitrate did provide NO for binding to superoxide, then peroxynitrite levels would also increase and studies have shown that chronic peroxyntrite causes vasodilation, but this quickly becomes tachyphylactic and loses potency or even increases BP and renal injury. The authors may want to address this possibility in their Discussion.

Author Response: We thank Reviewer #2 for making this important point. To address this question, we investigated the expression of nitrotyrosine as a surrogate marker of oxidative stress in mouse kidney and aorta samples by immunohistochemistry. Nitrotyrosine provides evidence of the generation of reactive nitrogen species, and is generated when peroxynitrite (ONOO-) is added to tyrosine itself or proteins containing tyrosine residues [18]. 

• No differences in nitrotyrosine expression were observed in the kidney or aorta between wild-type or Pkd1RC/RC mice, regardless of NaNO3 supplementation.

• Therefore, chronic nitrate supplementation had no effect on oxidative stress in this model. In contrast, previous studies have demonstrated that potassium nitrate supplementation led to lower plasma nitrotyrosine levels [19, 20]. Therefore, we agree with the Reviewer that the inefficacy of long-term oral nitrate supplementation in this study may have also been because oxidative stress was not altered.

• This data has been added to the Materials and methods (pg. 12, Line 222), Results (pg. 23, Lines 452-69 and Fig 11) and the Discussion has been modified accordingly (p25, Lines 500-09). 

Reviewer #3: 

Reviewer #3 Comment 1: To test their hypothesis of oral nitrate, to reduce cardiovascular and kidney diseases progression, authors didn’t establish reliable control before executing their animal model. experiments. 

Author Response: Thank you for raising this point. 

• The differences in cardiovascular phenotype between wild-type and Pkd1RC/RC mice is mild. We have now added this as a limitation of our study (see Discussion: p27-28, Lines 559-67), and further studies in other models of PKD are warranted. 

• It is important to note that there are no genetically orthologous models of ADPKD that exhibit measurable long-term cardiovascular disease progression, as vascular defects due to PKD1 and/or PKD2 mutations often lead to embryonic lethality [1-4]. Other experimental models of PKD that exhibit worsening cardiovascular disease over a measurable experimental time period are not genetically orthologous [5, 6], and thus translatability of this intervention remains uncertain even if oral nitrate supplementation is shown to be beneficial in these hypertensive PKD models. 

• The Pkd1RC/RC mouse model is the best validated preclinical model of ADPKD [21], and our characterisation of its cardiovascular phenotype will be very useful for future studies.

• Our animal study protocol was based on previous studies [10]. Several previous studies have added sodium nitrate (NaNO3) to drinking water to examine the efficacy of oral nitrate supplementation in murine disease models [10, 22]. Moreover, numerous previous studies have used sodium chloride as a control [10, 22]. 

Reviewer #3 Comment 2: Despite the fact that oral nitrate has been shown to reduce blood pressure (J Appl Physiol (1985). 2019 Oct 1;127(4):1085-1094.), Wild-type (control) treated with high dose of nitrate had an increase blood pressure (Figure 6 D-F). This raises the question of the purity and/or the nitrate supplement that was used in this study. 

Author Response: Thank you for raising this important question. 

• With regard to the question of purity, we have included more details in the manuscript to explain this further (see Materials and Methods: p9, Lines 151-52 and Lines 154-56; p13, Lines 254-55 and Results: p18, Lines 345-48 and S2 Table):

- First, the sodium nitrate used in this study was purchased from Sigma-Aldrich and had ≥99.0% purity. It was also dissolved in ultrapure water (MilliQ®), removing any possibility of other contaminants. 

- Second, the supplemented drinking water was prepared fresh and replaced every 7 days, and we also confirmed the stability of the nitrate and nitrite levels in the drinking water over this time period (see S2 Table).

- Finally, nitrate and nitrite levels in the supplemented drinking water were measured by an independent commercial analytical water laboratory (see S2 Table), and confirmed nitrate was present in the sodium nitrate supplemented drinking water, and absent in the ultrapure water itself or the sodium chloride supplemented drinking water. 

• With regard to the blood pressure data, please our response to Reviewer #1 Comment 2 for an explanation.

Reviewer #3 Comment 3: Or, perhaps the (PKD1 RC/RC mouse model) have an impaired microbiota in bioactivation of dietary nitrate (Free Radic Biol Med. 2019 Dec;145:342-348.).

Author Response: Thank you for raising this important point and also for providing the reference by Moretti et al.

• We agree with the Reviewer that both oral and gut microbiota are vital for the efficacy of the nitrate-nitrite-NO pathway [23-26]. 

• Increasing evidence suggests that variation in the gut microbiota exists in hypertension and kidney diseases including chronic kidney disease and acute kidney injury [27]. Though little is known about the gut microbiome of patients with ADPKD, a recent pilot study revealed that there were changes in gut microbiota in ADPKD patients according to renal function [28]. 

• It remains unclear whether there are disease-specific variations in the oral and gut microbiome of ADPKD patients, and it is well known that microbiota is influenced by age, genetics and diet [29, 30]. 

• Moreover, there may also be species- and/or strain-specific differences in the microbiome of mice and humans. Mice are not able to fully recapitulate human microbiome systems as: (1) inbred mouse strains do not represent the genetic variations in the human population, (2) housing conditions are strictly controlled, and (3) mice are fed a standardized chow diet [31].

• These points have been added to the revised manuscript (see Discussion: p27, Lines 549-57). 

Reviewer #3 Comment 4: Page 14: Figure 1 need to be mentioned in the result section before start interpreting Figure 2.

Author Response: This has been corrected. All main figures are mentioned in the Results in order. 

Reviewer #3 Comment 5: Page 15, Line 282: Please delete the duplication: “P<0.05 compared to compared to…”

Author Response: This has been corrected. 

Reviewer #3 Comment 6: Figure 3 and Figure 4: kidney tissue Images are very poor quality that is not easy to asses.

Author Response: We checked the image quality and noted the PDF proof is compressed. The images uploaded to the PLOS One server are of high quality. 

Reviewer #3 Comment 7: Figure 4 E-F: It is strange that the authors have examined the expression of iNOS (4 c-d) and kind of ignored all together the expression of eNOS to validate their immunohistochemistry results.

Author Response: 

• Western blot for eNOS was attempted in mouse kidney tissue, but unfortunately did not produce specific bands. As shown in the immunohistochemistry data (Fig 2E in the revised manuscript), eNOS in the mouse kidney was most strongly expressed in endothelial cells which are in low abundance in whole kidney tissue. The eNOS western blot data in kidney tissue has been added as a Supplemental Figure (S2 Fig) and a comment has been added to Results (p15, Lines 291-93). 

Reviewer #3 Comment 8: Flow of the figures and results is hard to follow, and need be to more organize. For example, Supplemental figures are not in order throughout the manuscript (S1 vs S2 and so on).

Author Response: The flow of the figures and results has been organized. 

Reviewer #3 Comment 9: Table 1: There is no rational, or nothing was given by authors, is to why they selected MCDK cell line over WT9-7 and/or WT9-12, nor any justification of selected concentrations.

Author Response: Justification for the 3D MDCK cyst model, WT cell lines and concentrations used is now provided in the revised manuscript. 

• The 3D MDCK cyst model is commonly used in ADPKD research [32]. In the present study, the model was used to screen for the efficacy of NO modulation on in vitro cyst growth (see Materials and methods: p8, Lines 123-24). 

• The WT 9-7 and WT 9-12 cells allowed us to examine NO expression in human cells possessing a mutated PKD1 variant, but these cells are not readily able to be grown in 3D culture. 

• The concentrations of L-arginine, sodium nitroprusside and L-NAME were selected based on previous studies performed on MDCK cells [33]. This reference has been added to the manuscript (see Materials and methods: p8, Lines 132-133). 

Reviewer #3 Comment 10: Table 1: Page 17, Line 329: Again please delete the duplication “P<0.05 compared to compared to…”

Author Response: This has been corrected. 

Reviewer #3 Comment 11: Table 1: Figure 5: Again, inconsistency of representing the results. Kidney weight and kidney to body weight ratio are given, but not the same with heart.

Author Response: Thank you for the suggestion. We have now added heart weight to the main figure (see Fig. 4E in the revised manuscript; previously Fig 5). 

Reviewer #3 Comment 12: Also, authors in (Fig 5C-E) are over interpreting these data, Page 17, Line 336, “Continuous treatment with NaNO3 for 8 months…did not alter the progression of kidney or heart enlargement in Pkd1RC/RC mice.” For example, in figure 5C low dose of Nitrate seems to have directly reduced the kidney enlargement, despite the KW: BW. In the contrary, moderate dose of Nitrate had at least increased heart to body weight ratio. This further might suggest that Nitrate effect might be dose dependent. Have the authors at least sub-analyzed these data by gender?

Author Response: Thank you for highlighting these findings. 

• Two kidney weight of Pkd1RC/RC mice on low dose nitrate supplementation was not different to wild-type control [P=0.087; Fig 4C in the revised manuscript]. However, it was also not different compared to Pkd1RC/RC control (P=0.182). 

• HW: BW ratio in Pkd1RC/RC mice in moderate dose nitrate supplementation was higher compared to wild-type control [P=0.023; Fig 4F in the revised manuscript]. However, similarly, it was not different compared to Pkd1RC/RC control (P=0.952). 

• Therefore, based on KW:BW ratio and HW:BW ratio and compared to type-matched controls, kidney and heart enlargement was not altered by nitrate supplementation i.e. no differences between wild-type high dose nitrate vs. wild-type control and no differences between nitrate-supplemented Pkd1RC/RC mice compared to Pkd1RC/RC control. 

• The line in the Results (p17, Lines 324-27) has now been modified to: “Continuous treatment with NaNO¬3 for 8 months…did not alter the progression of kidney or heart enlargement in Pkd1RC/RC mice compared to Pkd1RC/RC control (Fig 4C-F).”

• Sub-analysis by gender is provided in the Supplemental Tables (see S1 Table) and no differences were observed (see Results: p17, Lines 327-28). 

Reviewer #3 Comment 13: Table 1: Page 17, Line 363: confusing statement and perhaps incorrect. “Neither tail-cuff blood pressure nor heart rate were altered by long-term NaNO3 supplementation in wild-type or Pkd1RC/RC mice (Fig 6A-I & S3A-C Fig).” First, Blood pressure of Wild-type had indeed been negatively altered by long-term NaNO3 (Figure 6 D-F), particularly in the 6th month. Second, again authors need to explain that how high dose of nitrate is increasing BP rather than decreasing it in their control. This raises the questions of the purity and/or the supplement that was administered to the mice. 

Author Response: 

• We agree with Reviewer #3 and have added this result to the manuscript (Results: p19, Lines 364-68). Please see our response to Reviewer #1 Comment 2 for explanations for higher blood pressure observed in wild-type mice receiving high dose nitrate supplementation.

• The purity of the supplement was confirmed by an independent commercial laboratory (Sydney Analytical Laboratories, Seven Hills, NSW Australia) and we confirmed that the nitrate content of the drinking water was consistent over 7 days (see Materials and Methods: p9, Lines 151-52 and Lines 154-56; p13, Lines 254-55 and Results: p18, Lines 345-48 and S2 Table). 

Reviewer #3 Comment 14: Further…please move more relevant heart rate supplemental result to the main figure.

Author Response: Thank you for the suggestion. The heart rate data has been moved from Supplemental Figures to the main figures (see Fig 5D, G & K in the revised manuscript). 

Journal Requirements:

Please ensure that your manuscript meets PLOS ONE's style requirements, including those for file naming. PLOS ONE now requires that authors provide the original uncropped and unadjusted images underlying all blot or gel results reported in a submission’s figures or Supporting Information files. 

Author Response: These have been added to revised manuscript. The uncropped and unadjusted images for blots are shown in Supplemental Figs S2C-D and S3. 

“data not shown”; We require that authors provide all relevant data within the paper, Supporting Information files, or in an acceptable, public repository. Or, if the data are not a core part of the research being presented in your study, we ask that you remove the phrase that refers to these data.

Author Response: All data is now presented within the paper and the phrase “data not shown” has been removed.

REFERENCES

1. Kim K, Drummond I, Ibraghimov-Beskrovnaya O, Klinger K, Arnaout MA. Polycystin 1 is required for the structural integrity of blood vessels. Proceedings of the National Academy of Sciences. 2000;97(4):1731. doi: 10.1073/pnas.040550097.

2. Wu G, Markowitz GS, Li L, D'Agati VD, Factor SM, Geng L, et al. Cardiac defects and renal failure in mice with targeted mutations in Pkd2. Nat Genet. 2000;24(1):75-8. Epub 1999/12/30. doi: 10.1038/71724. PubMed PMID: 10615132.

3. Boulter C, Mulroy S, Webb S, Fleming S, Brindle K, Sandford R. Cardiovascular, skeletal, and renal defects in mice with a targeted disruption of the Pkd1 gene. Proc Natl Acad Sci U S A. 2001;98(21):12174-9. Epub 2001/10/11. doi: 10.1073/pnas.211191098. PubMed PMID: 11593033; PubMed Central PMCID: PMCPMC59787.

4. Muto S, Aiba A, Saito Y, Nakao K, Nakamura K, Tomita K, et al. Pioglitazone improves the phenotype and molecular defects of a targeted Pkd1 mutant. Hum Mol Genet. 2002;11(15):1731-42. Epub 2002/07/04. doi: 10.1093/hmg/11.15.1731. PubMed PMID: 12095915.

5. Kaspareit-Rittinghausen J, Deerberg F, Rapp KG, Wcislo A. Renal hypertension in rats with hereditary polycystic kidney disease. Z Versuchstierkd. 1990;33(5):201-4. Epub 1990/01/01. PubMed PMID: 2267865.

6. Phillips JK, Hopwood D, Loxley RA, Ghatora K, Coombes JD, Tan YS, et al. Temporal Relationship between Renal Cyst Development, Hypertension and Cardiac Hypertrophy in a New Rat Model of Autosomal Recessive Polycystic Kidney Disease. Kidney and Blood Pressure Research. 2007;30(3):129-44. doi: 10.1159/000101828.

7. Carlstrom M, Liu M, Yang T, Zollbrecht C, Huang L, Peleli M, et al. Cross-talk Between Nitrate-Nitrite-NO and NO Synthase Pathways in Control of Vascular NO Homeostasis. Antioxid Redox Signal. 2015;23(4):295-306. Epub 2013/11/15. doi: 10.1089/ars.2013.5481. PubMed PMID: 24224525; PubMed Central PMCID: PMCPMC4523008.

8. Villa LM, Salas E, Darley-Usmar VM, Radomski MW, Moncada S. Peroxynitrite induces both vasodilatation and impaired vascular relaxation in the isolated perfused rat heart. Proc Natl Acad Sci U S A. 1994;91(26):12383-7. doi: 10.1073/pnas.91.26.12383. PubMed PMID: 7809045.

9. Matavelli LC, Wells JE, Castillo A, Majid DS. Chronic treatment with a peroxynitrite scavenger attenuates blood pressure and improves renal hemodynamics in angiotensin II (ANG II) induced hypertensive rats. The FASEB Journal. 2008;22(S1):1160.1-.1. doi: 10.1096/fasebj.22.1_supplement.1160.1.

10. Bakker JR, Bondonno NP, Gaspari TA, Kemp-Harper BK, McCashney AJ, Hodgson JM, et al. Low dose dietary nitrate improves endothelial dysfunction and plaque stability in the ApoE−/−mouse fed a high fat diet. Free Radic Biol Med. 2016;99:189-98. doi: 10.1016/j.freeradbiomed.2016.08.009.

11. Hord NG, Tang Y, Bryan NS. Food sources of nitrates and nitrites: the physiologic context for potential health benefits. Am J Clin Nutr. 2009;90(1):1-10. doi: 10.3945/ajcn.2008.27131.

12. Lidder S, Webb AJ. Vascular effects of dietary nitrate (as found in green leafy vegetables and beetroot) via the nitrate-nitrite-nitric oxide pathway. Br J Clin Pharmacol. 2013;75(3):677-96. Epub 2012/08/14. doi: 10.1111/j.1365-2125.2012.04420.x. PubMed PMID: 22882425; PubMed Central PMCID: PMCPMC3575935.

13. Lundberg JO, Carlström M, Weitzberg E. Metabolic Effects of Dietary Nitrate in Health and Disease. Cell Metab. 2018;28(1):9-22. Epub 2018/07/05. doi: 10.1016/j.cmet.2018.06.007. PubMed PMID: 29972800.

14. Gürtler A, Kunz N, Gomolka M, Hornhardt S, Friedl AA, McDonald K, et al. Stain-Free technology as a normalization tool in Western blot analysis. Analytical biochemistry. 2013;433(2):105-11. Epub 2012/10/23. doi: 10.1016/j.ab.2012.10.010. PubMed PMID: 23085117.

15. Taylor SC, Berkelman T, Yadav G, Hammond M. A defined methodology for reliable quantification of Western blot data. Mol Biotechnol. 2013;55(3):217-26. Epub 2013/05/28. doi: 10.1007/s12033-013-9672-6. PubMed PMID: 23709336; PubMed Central PMCID: PMCPMC3840294.

16. Csonka C, Páli T, Bencsik P, Görbe A, Ferdinandy P, Csont T. Measurement of NO in biological samples. Br J Pharmacol. 2015;172(6):1620-32. Epub 2014/07/06. doi: 10.1111/bph.12832. PubMed PMID: 24990201; PubMed Central PMCID: PMCPMC4369268.

17. Bryan NS, Grisham MB. Methods to detect nitric oxide and its metabolites in biological samples. Free radical biology & medicine. 2007;43(5):645-57. Epub 2007/04/29. doi: 10.1016/j.freeradbiomed.2007.04.026. PubMed PMID: 17664129.

18. Halliwell B. What nitrates tyrosine? Is nitrotyrosine specific as a biomarker of peroxynitrite formation in vivo? FEBS Lett. 1997;411(2):157-60. doi: 10.1016/S0014-5793(97)00469-9.

19. Ashor AW, Shannon OM, Werner AD, Scialo F, Gilliard CN, Cassel KS, et al. Effects of inorganic nitrate and vitamin C co-supplementation on blood pressure and vascular function in younger and older healthy adults: A randomised double-blind crossover trial. Clin Nutr. 2020;39(3):708-17. Epub 2019/04/02. doi: 10.1016/j.clnu.2019.03.006. PubMed PMID: 30930132.

20. Ashor AW, Chowdhury S, Oggioni C, Qadir O, Brandt K, Ishaq A, et al. Inorganic Nitrate Supplementation in Young and Old Obese Adults Does Not Affect Acute Glucose and Insulin Responses but Lowers Oxidative Stress. The Journal of Nutrition. 2016;146(11):2224-32. doi: 10.3945/jn.116.237529.

21. Hopp K, Ward CJ, Hommerding CJ, Nasr SH, Tuan HF, Gainullin VG, et al. Functional polycystin-1 dosage governs autosomal dominant polycystic kidney disease severity. J Clin Invest. 2012;122(11):4257-73. Epub 2012/10/16. doi: 10.1172/jci64313. PubMed PMID: 23064367; PubMed Central PMCID: PMCPMC3484456.

22. Roberts LD, Ashmore T, Kotwica AO, Murfitt SA, Fernandez BO, Feelisch M, et al. Inorganic nitrate promotes the browning of white adipose tissue through the nitrate-nitrite-nitric oxide pathway. Diabetes. 2015;64(2):471-84. doi: 10.2337/db14-0496.

23. Tiso M, Schechter AN. Nitrate reduction to nitrite, nitric oxide and ammonia by gut bacteria under physiological conditions. PLOS One. 2015;10(3):e0119712-e. doi: 10.1371/journal.pone.0119712. PubMed PMID: 25803049.

24. Goh CE, Trinh P, Colombo PC, Genkinger JM, Mathema B, Uhlemann AC, et al. Association Between Nitrate-Reducing Oral Bacteria and Cardiometabolic Outcomes: Results From ORIGINS. J Am Heart Assoc. 2019;8(23):e013324. Epub 2019/11/27. doi: 10.1161/jaha.119.013324. PubMed PMID: 31766976; PubMed Central PMCID: PMCPMC6912959.

25. Bondonno CP, Liu AH, Croft KD, Considine MJ, Puddey IB, Woodman RJ, et al. Antibacterial Mouthwash Blunts Oral Nitrate Reduction and Increases Blood Pressure in Treated Hypertensive Men and Women. Am J Hypertens. 2015;28(5):572-5. doi: 10.1093/ajh/hpu192.

26. Moretti C, Zhuge Z, Zhang G, Haworth SM, Paulo LL, Guimarães DD, et al. The obligatory role of host microbiota in bioactivation of dietary nitrate. Free Radic Biol Med. 2019;145:342-8. doi: 10.1016/j.freeradbiomed.2019.10.003.

27. Chen Y-Y, Chen D-Q, Chen L, Liu J-R, Vaziri ND, Guo Y, et al. Microbiome–metabolome reveals the contribution of gut–kidney axis on kidney disease. J Transl Med. 2019;17(1):5. doi: 10.1186/s12967-018-1756-4.

28. Yacoub R, Nadkarni GN, McSkimming DI, Chaves LD, Abyad S, Bryniarski MA, et al. Fecal microbiota analysis of polycystic kidney disease patients according to renal function: A pilot study. Exp Biol Med (Maywood). 2019;244(6):505-13. Epub 2018/12/13. doi: 10.1177/1535370218818175. PubMed PMID: 30539656; PubMed Central PMCID: PMCPMC6547006.

29. Rosier BT, Marsh PD, Mira A. Resilience of the Oral Microbiota in Health: Mechanisms That Prevent Dysbiosis. J Dent Res. 2018;97(4):371-80. Epub 2017/12/02. doi: 10.1177/0022034517742139. PubMed PMID: 29195050.

30. Wen L, Duffy A. Factors Influencing the Gut Microbiota, Inflammation, and Type 2 Diabetes. The Journal of nutrition. 2017;147(7):1468S-75S. Epub 2017/06/14. doi: 10.3945/jn.116.240754. PubMed PMID: 28615382.

31. Nguyen TLA, Vieira-Silva S, Liston A, Raes J. How informative is the mouse for human gut microbiota research? Disease Models & Mechanisms. 2015;8(1):1. doi: 10.1242/dmm.017400.

32. Turner CM, King BF, Srai KS, Unwin RJ. Antagonism of endogenous putative P2Y receptors reduces the growth of MDCK-derived cysts cultured in vitro. Am J Physiol Renal Physiol. 2007;292(1):F15-F25. Epub 2006/07/18. doi: 10.1152/ajprenal.00103.2006. PubMed PMID: 16849696.

33. Mazzei L, García IM, Cacciamani V, Benardón ME, Manucha W. WT-1 mRNA expression is modulated by nitric oxide availability and Hsp70 interaction after neonatal unilateral ureteral obstruction. Biocell. 2010;34(3):121-32. Epub 2011/03/30. PubMed PMID: 21443142.

---

## [Decision Letter · Decision Letter 1]

7 Jan 2021

PONE-D-20-29428R1

Effects of long-term oral nitrate supplementation on the progression of cardiovascular and renal outcomes in murine autosomal dominant polycystic kidney disease

PLOS ONE

Dear Dr. Zhang,

Thank you for submitting your manuscript to PLOS ONE. After careful consideration, we feel that it has merit but does not fully meet PLOS ONE’s publication criteria as it currently stands. Therefore, we invite you to submit a revised version of the manuscript that addresses the points raised during the review process.

Fortunately, reviewers #2 and #3 are satisfied with your revision. However, reviewer #1 still has several issues that you need to address. They all relate to the validity of your conclusions with respect to ADPKD considering 1) the mild phenotype of your model and 2) your extrapolation from findings in a cultured cell model. Thus your conclusions relating to in vivo effects of nitrates in ADPKD need to be (further) toned down. Reviewer #1 also still finds the manuscript somewhat chaotic and I agree. The manuscript will have to be (partly) restructured. Please present ALL in vitro data first and then present your study characterising NOS expression in untreated ADPKD and wild-type mice. Next, use ALL these data to motivate your intervention study. Most of the figures relating to your intervention study show no effects of the intervention and can be moved to the supplement. More emphasis in the discussion should be on the in vitro findings and less on the intervention study.

We look forward to receiving your revised manuscript.

Kind regards,

Jaap A. Joles, DVM, PhD

Academic Editor

PLOS ONE

Reviewers' comments:

Reviewer's Responses to Questions

**Comments to the Author**

1. If the authors have adequately addressed your comments raised in a previous round of review and you feel that this manuscript is now acceptable for publication, you may indicate that here to bypass the “Comments to the Author” section, enter your conflict of interest statement in the “Confidential to Editor” section, and submit your "Accept" recommendation.

Reviewer #1: (No Response)

Reviewer #2: All comments have been addressed

Reviewer #3: All comments have been addressed

2. Is the manuscript technically sound, and do the data support the conclusions?

Reviewer #1: No

Reviewer #2: Yes

Reviewer #3: Yes

3. Has the statistical analysis been performed appropriately and rigorously? 

Reviewer #1: Yes

Reviewer #2: Yes

Reviewer #3: Yes

4. Have the authors made all data underlying the findings in their manuscript fully available?

Reviewer #1: Yes

Reviewer #2: Yes

Reviewer #3: Yes

5. Is the manuscript presented in an intelligible fashion and written in standard English?

Reviewer #1: Yes

Reviewer #2: Yes

Reviewer #3: Yes

6. Review Comments to the Author

Reviewer #1: In this study, Dr. Rangan investigate the effect of nitrates in a mouse model of ADPKD. They observed that oral nitrates do not change the cardiovascular profile in controls or PKD animals, and at the same time do not reduce the cystic changes in PKD. I appreciate the comments from the investigators on the prior review.

However, as in the prior review, my main concern is that there are no changes in the cardiovascular phenotype between PKD and control. Authors postulate that this is a very mild phenotype. The way the data is presented, there are no cardiovascular changes between wild type and PKD. On some level, the conclusion that there are no effects on cardiovascular phenotype is true, however I do not think it should be concluded that nitrates do not affect the cardiovascular outcomes (we do not see any on the and treated group) in PKD.

To my read, the change in nitric oxide is supported by the DAF data (cell culture of human cells) and is not related to the ADPKD model. If that is the case, the conclusion that there is a decrease in nitric oxide in the RCRC murine model of PKD is not supported. Please clarify if I am missing something.

In addition, I feel that too much data is presented together, cystic renal changes together with cardiovascular changes that are not truly there and the paper altogether appears to diffuse. Consider focusing on the cystic changes.

Other comments:

1. The data on the renal tubular cell line is not from the specific PKD model mentioned in the title and is somewhat out of place.

2. Figure 1. What is the effect of nitrates on the cell line?

Reviewer #2: The authors have adequately addressed my concerns. Required questions have been answered and that all responses meet formatting specifications.

Reviewer #3: (No Response)

7. PLOS authors have the option to publish the peer review history of their article (what does this mean?). If published, this will include your full peer review and any attached files.

Reviewer #1: No

Reviewer #2: No

Reviewer #3: No

---

## [Author Response · Author response to Decision Letter 1]

22 Jan 2021

Response to Editor’s Comments

Editor Comment #1: Reviewer #1 still has several issues that you need to address. They all relate to the validity of your conclusions with respect to ADPKD considering 1) the mild phenotype of your model and 2) your extrapolation from findings in a cultured cell model. Your conclusions relating to in vivo effects of nitrates in ADPKD need to be (further) toned down. 

Response: Thank you for the opportunity to further improve this manuscript. We agree and have extensively revised the entire manuscript. For the in vivo study in the Pkd1RC/RC mice, we have now placed all emphasis on the efficacy of nitrates on renal cyst growth. Furthermore, due to the lack of cardiovascular disease phenotype in the Pkd1RC/RC mouse, we agree, and have stated throughout the manuscript, that it is not possible to conclude whether nitrates have efficacy on this complication (see Abstract: p3, Lines 43-46; Results: p18, Lines 377-380; and Discussion: p23-24, Lines 495-497).

Editor Comment #2: Reviewer #1 also still finds the manuscript somewhat chaotic and I agree. The manuscript will have to be (partly) restructured. Please present ALL in vitro data first and then present your study characterising NOS expression in untreated ADPKD and wild-type mice. Next, use ALL these data to motivate your intervention study. Most of the figures relating to your intervention study show no effects of the intervention and can be moved to the supplement. 

Response: We agree, and the entire manuscript has now been extensively re-written and edited, and the Results have also been partly re-structured. As suggested, all in vitro data is presented first followed by in vivo data. Figures related to cardiovascular phenotype have moved to the Supporting Information (see Figs S6-8).

Editor Comment #3: More emphasis in the discussion should be on the in vitro findings and less on the intervention study.

Response: We agree, and the Discussion has also been extensively edited. More detailed discussion is provided on the in vitro findings (see Discussion: p21-22, Lines 428-460), and we have reduced the sections on the intervention study (see Discussion: p22-23, Lines 470-485).

Response to Reviewer #1 Comments

Reviewer Comment #1: My main concern is that there are no changes in the cardiovascular phenotype between PKD and control. Authors postulate that this is a very mild phenotype. The way the data is presented, there are no cardiovascular changes between wild type and PKD. On some level, the conclusion that there are no effects on cardiovascular phenotype is true, however I do not think it should be concluded that nitrates do not affect the cardiovascular outcomes (we do not see any on the and treated group) in PKD. 

Response: We thank Reviewer #1 for emphasising this point and we agree. Consequently, we have extensively revised the entire manuscript. Specifically, for the in vivo data in the Pkd1RC/RC mouse model, we placed all emphasis on the efficacy of nitrates on renal cyst growth. Furthermore, , due to the lack of cardiovascular disease phenotype in the Pkd1RC/RC mouse model, we have explicitly stated that it is not possible to conclude whether nitrates have efficacy on this complication (see Abstract: p3, Lines 43-46; Results: p18, Lines 377-380; and Discussion: p23-24, Lines 495-497).

Reviewer Comment #2: To my read, the change in nitric oxide is supported by the DAF data (cell culture of human cells) and is not related to the ADPKD model. If that is the case, the conclusion that there is a decrease in nitric oxide in the RCRC murine model of PKD is not supported. Please clarify if I am missing something.

Response: We thank Reviewer #1 for this comment. The manuscript has been revised to emphasise the relevance of the human ADPKD cell line data as well as the in vitro cyst model (see Introduction: p5, Lines 88-91; Materials and Methods: p7, Lines 125-127; Discussion p21-22, Lines 428-460). Furthermore, urinary nitrate excretion in PKD1 mice was reduced compared to wild-type mice (see Results: p16, Lines 326-327; Table 2), suggesting that nitric oxide is attenuated in this model, and we have highlighted this data, as it was not discussed in the previous version.

Reviewer Comment #3: I feel that too much data is presented together, cystic renal changes together with cardiovascular changes that are not truly there and the paper altogether appears to diffuse. Consider focusing on the cystic changes.

Response: We agree, and the entire manuscript has now been extensively re-written and edited, and the Results have also been partly re-structured. For the in vivo study in the Pkd1RC/RC mice, we have now placed all emphasis on the efficacy of nitrates on renal cyst growth. The Results text related to cardiovascular disease outcomes has been significantly condensed and figures have moved to the Supporting Information (see Results: p18, Lines 377-380; Figs S6-8).

Reviewer Comment #4: The data on the renal tubular cell line is not from the specific PKD model mentioned in the title and is somewhat out of place.

Response: We have enhanced our explanation of the rationale for studies conducted in the ADPKD cell lines (see Introduction: p5, Lines 88-91), and also revised the title to refer to “experimental ADPKD” so that the clarity of studies performed is improved.

Reviewer Comment #5: Figure 1. What is the effect of nitrates on the cell line?

Response: This is an interesting point and was also raised by the research team during the planning of the study. We did not investigate this hypothesis because conversion of nitrate to NO via the entero-salivary pathway is dependent on the presence of oral and gut bacteria [1-4] (see Discussion: p23, 488-492). Therefore, it would not be possible to easily replicate this process in our simple single-cell based cell culture model. However, it would interesting to consider this in future studies, should oral nitrates be shown to have efficacy on cardiovascular disease in other PKD models. In any case, our in vivo studies demonstrated that there was no efficacy on kidney cyst growth, and we feel that this is the most important finding. 

References 

1. Tiso M, Schechter AN. Nitrate reduction to nitrite, nitric oxide and ammonia by gut bacteria under physiological conditions. PLOS One. 2015;10(3):e0119712-e. doi: 10.1371/journal.pone.0119712. PubMed PMID: 25803049.

2. Bondonno CP, Liu AH, Croft KD, Considine MJ, Puddey IB, Woodman RJ, et al. Antibacterial Mouthwash Blunts Oral Nitrate Reduction and Increases Blood Pressure in Treated Hypertensive Men and Women. Am J Hypertens. 2015;28(5):572-5. doi: 10.1093/ajh/hpu192.

3. Goh CE, Trinh P, Colombo PC, Genkinger JM, Mathema B, Uhlemann AC, et al. Association Between Nitrate-Reducing Oral Bacteria and Cardiometabolic Outcomes: Results From ORIGINS. J Am Heart Assoc. 2019;8(23):e013324. Epub 2019/11/27. doi: 10.1161/jaha.119.013324. PubMed PMID: 31766976; PubMed Central PMCID: PMCPMC6912959.

4. Moretti C, Zhuge Z, Zhang G, Haworth SM, Paulo LL, Guimarães DD, et al. The obligatory role of host microbiota in bioactivation of dietary nitrate. Free Radic Biol Med. 2019;145:342-8. doi: 10.1016/j.freeradbiomed.2019.10.003.

---

## [Decision Letter · Decision Letter 2]

2 Feb 2021

PONE-D-20-29428R2

Long-term dietary nitrate supplementation does not reduce renal cyst growth in experimental autosomal dominant polycystic kidney disease

PLOS ONE

Dear Dr. Zhang,

Thank you for submitting your manuscript to PLOS ONE. After careful consideration, we feel that it has merit but does not fully meet PLOS ONE’s publication criteria as it currently stands. Therefore, we invite you to submit a revised version of the manuscript that addresses the points raised during the review process.

More changes are required, particularly in the abstract. You really must take the cardiovascular aspect out of the paper to merit acceptance. Make sure that you adequately address all the remaining comments.

We look forward to receiving your revised manuscript.

Kind regards,

Jaap A. Joles, DVM, PhD

Academic Editor

PLOS ONE

Reviewers' comments:

Reviewer's Responses to Questions

**Comments to the Author**

1. If the authors have adequately addressed your comments raised in a previous round of review and you feel that this manuscript is now acceptable for publication, you may indicate that here to bypass the “Comments to the Author” section, enter your conflict of interest statement in the “Confidential to Editor” section, and submit your "Accept" recommendation.

Reviewer #1: (No Response)

2. Is the manuscript technically sound, and do the data support the conclusions?

Reviewer #1: Partly

3. Has the statistical analysis been performed appropriately and rigorously? 

Reviewer #1: I Don't Know

4. Have the authors made all data underlying the findings in their manuscript fully available?

Reviewer #1: Yes

5. Is the manuscript presented in an intelligible fashion and written in standard English?

Reviewer #1: Yes

6. Review Comments to the Author

Reviewer #1: In this study, Zhang et al study the role of nitrate supplementation on renal cyst growth in ADPKD (RCRC model). They observed that 3 different doses of nitrates over 8 months did not affect cyst growth. Furthermore they did not observe any cardiovascular changes between wt and ADPKD, therefore no conclusion can be extracted regarding the effect of nitrates.

I appreciate the significant changes that the authors have introduced in the different version of the manuscript.

Comments:

1. The majority of the introduction has to do with endothelial dysfunction and vascular changes but again is not the focus of the paper in this version (neither on the title nor the results section). In other words I do not see how the introduction leads to the test of nitrates on cyst growth.

2. I appreciate the toning down of the cardiovascular data in this version. However, I do not think that they have a role in this manuscript. I would recommend removal of the sentence in the abstract starting with “the phenotype of cardiovascular disease…” as well as the myography and other CV data is not relevant here. BP may provide support for the mild nature of this model, however I would remove the use of nitrates on cardiovascular outcomes (there are none on the disease model).

3. Page 18. See comment 2.

4. Figure 6. I do not see any quantification differences. If there are some on the staining, they should be made more clear the graphics.

5. Could be that the decrease in urinary nitrate reflect more than just a decrease in NO bioavailability? Renal function? It is good to see that the animal that receive nitrates have higher levels in the urine, demonstrating that the actually received the nitrates. I am not sure that this piece of data reinforces the hypothesis. Please clarify if I am missing something.

7. PLOS authors have the option to publish the peer review history of their article (what does this mean?). If published, this will include your full peer review and any attached files.

Reviewer #1: No

---

## [Author Response · Author response to Decision Letter 2]

18 Feb 2021

Response to Editor’s Comments

Editor Comment #1. More changes are required, particularly in the abstract. You really must take the cardiovascular aspect out of the paper to merit acceptance. Make sure that you adequately address all the remaining comments.

Author Response: We have removed the cardiovascular aspect of the data (aortic myography) from all the sections (See Abstract p3. Lines 43-46, Introduction p4. Lines 62-67, Methods p9. Lines 179-180; pp12-13, Lines 243-277, Results p18, Lines 377-380, Discussion pp.23-24 Lines 495-497; Supplemental Figures S1 and S7). 

Response to Reviewer’s Comments

Reviewer Comment #1. The majority of the introduction has to do with endothelial dysfunction and vascular changes but again is not the focus of the paper in this version (neither on the title nor the results section). In other words I do not see how the introduction leads to the test of nitrates on cyst growth.

Author Response: We have removed the section of the Introduction from p4 (Lines 62-67) where the endothelial dysfunction in ADPKD was discussed.

Reviewer Comment #2. I appreciate the toning down of the cardiovascular data in this version. However, I do not think that they have a role in this manuscript. I would recommend removal of the sentence in the abstract starting with “the phenotype of cardiovascular disease…” as well as the myography and other CV data is not relevant here. BP may provide support for the mild nature of this model, however I would remove the use of nitrates on cardiovascular outcomes (there are none on the disease model).

Author Response: We have removed the cardiovascular data from all the sections of the manuscript (See Abstract p3. Lines 43-46, Introduction p4. Lines 62-67, Methods p9. Lines 179-180; pp12-13. Lines 243-277, Results p18. Lines 377-380, Discussion pp.23-24. Lines 495-497; Supplemental Figures S1, S6-S8).

Reviewer Comment #3. Page 18. See comment 2.

Author Response: The results on cardiovascular phenotype (p18. Lines 377-380) have been removed.

Reviewer Comment #4. Figure 6. I do not see any quantification differences. If there are some on the staining, they should be made more clear the graphics.

Author Response: There were no quantitative differences in either nitrotyrosine or γH2AX expression in the wildtype or Pkd1RC/RC mice treated with NaNO3 compared to vehicle (Panels B to D). 

Reviewer Comment #5. Could be that the decrease in urinary nitrate reflect more than just a decrease in NO bioavailability? Renal function? It is good to see that the animal that receive nitrates have higher levels in the urine, demonstrating that the actually received the nitrates. I am not sure that this piece of data reinforces the hypothesis. Please clarify if I am missing something.

Author Response: Thank you for raising this important question. The renal functions was a part of our initial manuscript (but removed as per reviewer suggestion following first revision). Briefly, this data showed that serum urea in Pkd1RC/RC mice was not different to wildtype mice. On contrary, urinary nitrate in the Pkd1RC/RC mice increased with increasing nitrate supplementation strengthening the fact that renal filtration of nitrates is normal in the Pkd1RC/RC mice, and the lower nitrate in urine of Pkd1RC/RC mice (~30% of wildtype) supports the hypothesis that NO metabolism is reduced in the Pkd1RC/RC model.

---

## [Decision Letter · Decision Letter 3]

24 Feb 2021

PONE-D-20-29428R3

Long-term dietary nitrate supplementation does not reduce renal cyst growth in experimental autosomal dominant polycystic kidney disease

PLOS ONE

Dear Dr. Rangan,

Thank you for submitting your manuscript to PLOS ONE. After careful consideration, we feel that it has merit but does not fully meet PLOS ONE’s publication criteria as it currently stands. Therefore, we invite you to submit a revised version of the manuscript that addresses the points raised during the review process.

Please adress ALL remaining comments of reviewer #1 and also delete text on cardiovascular disease (see below).

We look forward to receiving your revised manuscript.

Kind regards,

Jaap A. Joles, DVM, PhD

Academic Editor

PLOS ONE

Journal Requirements:

Additional Editor Comments (if provided):

Besides adressing the final comments of reviewer #1 please also delete text on cardiovascular disease from the following sections:

Keywords (line 48)

Aims (line 87)

Discussion, last sentence (lines 460-461)

Reviewers' comments:

Reviewer's Responses to Questions

**Comments to the Author**

1. If the authors have adequately addressed your comments raised in a previous round of review and you feel that this manuscript is now acceptable for publication, you may indicate that here to bypass the “Comments to the Author” section, enter your conflict of interest statement in the “Confidential to Editor” section, and submit your "Accept" recommendation.

Reviewer #1: (No Response)

2. Is the manuscript technically sound, and do the data support the conclusions?

Reviewer #1: Partly

3. Has the statistical analysis been performed appropriately and rigorously? 

Reviewer #1: Yes

4. Have the authors made all data underlying the findings in their manuscript fully available?

Reviewer #1: Yes

5. Is the manuscript presented in an intelligible fashion and written in standard English?

Reviewer #1: No

6. Review Comments to the Author

Reviewer #1: In the revised version of the manuscript, the authors have focused on the cell culture findings. Specifically that nitric oxide supplementation improves cyst growth in MDCK PKD cell line.

Furthermore, they observed that different doses of nitrites did not improve cyst size in a PKD 1 mouse model.

I think that this version is much more focused and a lot of the previous data (that was not contributing much) was put aside.

Comments:

Figure 1: Please remove the aortic myography reference

Figure 6: authors observed that there is no change in their evaluation of the nitric oxide system in vivo between control in PKD. This is relevant because if we want to transfer the knowledge from cell culture to the in vivo setting, we would expect some sort of correlation between the endo findings which is not present. The fact that that correlation is not present, makes the cell cultures-in vivo translation a bit less relevant. That being said, this reviewer recognizes that authors did not do the same evaluation in cell culture and in vivo and thus a direct comparison is difficult to make. This issue must be adequately discussed.

7. PLOS authors have the option to publish the peer review history of their article (what does this mean?). If published, this will include your full peer review and any attached files.

Reviewer #1: No

---

## [Author Response · Author response to Decision Letter 3]

25 Feb 2021

Response to Comments of the Editors and Reviewers 

Editorial Comments: 

Please address ALL remaining comments of reviewer #1 and also delete text on cardiovascular disease:

• Keywords (line 48)

• Aims (line 87)

• Discussion, last sentence (lines 460-461)

Authors Response: Thank you. All text referring to cardiovascular disease has been removed, and all of the Reviewers comments have been addressed. Please see section below for further details.

Response to Reviewer #1

Comment #1: Figure 1: Please remove the aortic myography reference

Author Response: Thank you. The reference to aortic myography has been removed from Figure 1.

Comment #2: Figure 6: authors observed that there is no change in their evaluation of the nitric oxide system in vivo between control in PKD. This is relevant because if we want to transfer the knowledge from cell culture to the in vivo setting, we would expect some sort of correlation between the endo findings which is not present. The fact that that correlation is not present, makes the cell cultures-in vivo translation a bit less relevant. That being said, this reviewer recognizes that authors did not do the same evaluation in cell culture and in vivo and thus a direct comparison is difficult to make. This issue must be adequately discussed.

Author Response: Thank you for the comment. The experiments in Figure 6 were performed to assess whether oral nitrate consumption caused any of the adverse effects known to occur with excess intake. The latter includes oxidative stress (in particular nitrotyrosine formation) and DNA damage (gamma-H2AX). Our data showed that nitrate supplementation did not cause these mutagenic effects and was not an explanation for the lack of efficacy on kidney cyst growth. To address Reviewer #1 comments:

• the results section has been modified to improve clarity of this data (See lines 351-355); and 

• that the differences between the design of the in vivo and in vitro models prevents direct comparison of the data between these experiments, has been discussed as a limitation of the current study (See lines 437-448).

---

## [Editor Report · Decision Letter 4]

26 Feb 2021

Long-term dietary nitrate supplementation does not reduce renal cyst growth in experimental autosomal dominant polycystic kidney disease

PONE-D-20-29428R4

Dear Dr. Rangan,

We’re pleased to inform you that your manuscript has been judged scientifically suitable for publication and will be formally accepted for publication once it meets all outstanding technical requirements.

Kind regards,

Jaap A. Joles, DVM, PhD

Academic Editor

PLOS ONE
---

## [Editor Report · Acceptance letter]

12 Apr 2021

PONE-D-20-29428R4 

Long-term dietary nitrate supplementation does not reduce renal cyst growth in experimental autosomal dominant polycystic kidney disease 

Dear Dr. Rangan:

I'm pleased to inform you that your manuscript has been deemed suitable for publication in PLOS ONE. Congratulations! Your manuscript is now with our production department. 

Kind regards, 

on behalf of

Dr. Jaap A. Joles 

Academic Editor

PLOS ONE